# Annihilation of Spurious Minima in Two-Layer ReLU Networks

**Yossi Arjevani**
The Hebrew University
yossi.arjevani@gmail.com

**Michael Field**
UC Santa Barbara
Mike.field@gmail.com

## Abstract

We study the optimization problem associated with fitting two-layer ReLU neural networks with respect to the squared loss, where labels are generated by a target network. Use is made of the rich symmetry structure to develop a novel set of tools for studying the mechanism by which over-parameterization annihilates spurious minima. Sharp analytic estimates are obtained for the loss and the Hessian spectrum at different minima, and it is proved that adding neurons can turn symmetric spurious minima into saddles; minima of lesser symmetry require more neurons. Using Cauchy's interlacing theorem, we prove the existence of descent directions in certain subspaces arising from the symmetry structure of the loss function. This analytic approach uses techniques, new to the field, from algebraic geometry, representation theory and symmetry breaking, and confirms rigorously the effectiveness of over-parameterization in making the associated loss landscape accessible to gradient-based methods. For a fixed number of neurons and inputs, the spectral results remain true under symmetry breaking perturbation of the target.

## 1 Introduction

An outstanding question in deep learning (DL) concerns the ability of simple gradient-based methods to successfully train neural networks despite the nonconvexity of the associated optimization problems. Indeed, nonconvex optimization landscapes may have spurious (i.e., non-global local) minima with large basins of attraction and this can cause a complete failure of these methods. Evidence suggests that this problem can be circumvented by the use of a large number of parameters in DL models. In view of the complexity exhibited by contemporary neural networks and the absence of suitable analytic tools, much recent research has focused on two-layer ReLU networks as a realistic starting point for a theoretical study [1, 2, 3, 4, 5, 6, 7]. The two-layer networks considered were typically of the form:

$$f(\boldsymbol{x}; W, \boldsymbol{\alpha}) \coloneqq \boldsymbol{\alpha}^\top \varphi(W\boldsymbol{x}), \quad W \in M(k, d), \, \boldsymbol{\alpha} \in \mathbb{R}^k, \tag{1}$$

where $\varphi(z) \coloneqq \max\{0, z\}$ is the ReLU function acting entrywise and $M(k, d)$ denotes the space of $k \times d$ matrices. In order to isolate the study of optimization-related obstructions due to nonconvexity from issues pertaining to the expressive power of two-layer networks, data has been often assumed to be fully realizable. This was further motivated by hardness results which indicated a strict barrier inherent to the explanatory power of distribution-free approaches operating in complete generality [8, 9, 10]. For the squared loss, the resulting, highly nonconvex, expected loss is

$$\mathcal{L}(W, \boldsymbol{\alpha}) \coloneqq \frac{1}{2}\mathbb{E}_{\boldsymbol{x} \sim \mathcal{D}}\left[\left(f(\boldsymbol{x}; W, \boldsymbol{\alpha}) - f(\boldsymbol{x}; V, \boldsymbol{\beta})\right)^2\right], \tag{2}$$

where $\mathcal{D}$ denotes a probability distribution over the input space, $W \in M(k, d)$, $\boldsymbol{\alpha} \in \mathbb{R}^k$ are the optimization variables, and $V \in M(d, d)$, $\boldsymbol{\beta} \in \mathbb{R}^d$ are fixed parameters.

36th Conference on Neural Information Processing Systems (NeurIPS 2022).

The choice of the $d$-variate normal Gaussian distribution for the input distribution has drawn a considerable interest, e.g., [11, 12, 13, 14, 15, 16, 17]. Empirically, it has been observed that as the number of neurons $k$ increases, the loss of models obtained using stochastic gradient descent (SGD) under Xavier initialization [18] decreases (see [9, 19]). The objective of this work is to study the mathematical mechanism behind this fundamental phenomenon. Using ideas based on *symmetry breaking* (see Section 2), we are able to employ techniques from representation theory and algebraic geometry to study how the loss landscape is transformed when the number of neurons is increased. A key feature of our approach is the use of *Puiseux series* in $d$ and path based techniques. For example, we may define the spectrum of the Hessian as Puiseux series for *real* values of $d$ and so determine the values of $d$ (typically not integers) where eigenvalues change sign. These methods allow the proof of powerful analytic results: in this paper, we derive sharp analytic estimates of the loss and the Hessian spectrum at local minima allowing us to analyze the mechanism whereby spurious minima are annihilated when the number of neurons $k$ is increased. Our contributions can be summarized by

**Theorem 1** *(Informal) We describe several infinite families of spurious minima, partly characterized by their symmetry, and prove that:*
- *Adding rather few neurons can transform symmetric spurious minima into saddles.*
- *The Hessian spectrum remains extremely skewed with $\Theta(d)$ eigenvalues growing linearly with $d$, and $\Theta(kd)$ eigenvalues being $\Theta(1)$.*
- *Increasing the number of neurons adds $\Theta(d)$ descent directions in a $\Theta(d)$-dimensional subspace dictated by the symmetry structure (i.e., the isotypic decomposition) of the loss function.*
- *The loss remains (essentially) unchanged.*

The formation of new decent directions allows gradient-based methods to escape spurious minima that exist in the original loss landscape ($k = d$), and so detect models of reduced loss. To indicate the subtly of the results given in Theorem 1, consider a family of spurious minima with symmetry $\Delta(S_{d-1} \times S_1)$ (the type II family described in Section 3). Assume $k = d$. Spurious minima occur for $d \geq 6$. Adding one neuron results in these minima occurring for $d \geq 8$ — not promising. Adding two neurons annihilates *all* these spurious minima and creates no new spurious minima. The mechanism is finite, cannot be inferred from a limiting case, and persists under forced symmetry breaking, and so applies beyond symmetric targets. If instead we consider the unique family of spurious minima with isotropy $\Delta S_d$ (see Section 3), we find that adding just one neuron annihilates these spurious minima. This occurs through the appearance of a multiplicity $(d-1)$-eigenvalue associated to the standard representation of $S_d$ on $\mathbb{R}^{d-1}$ (all other eigenvalues remain positive). More precisely, if $k = d$ there are 3 strictly positive Hessian eigenvalues associated to the standard representation of $S_d$, each with multiplicity $d - 1$. Modulo $O(d^{-\frac{1}{2}})$ terms, they are $\frac{1}{4} - \frac{1}{2\pi}$, $\frac{1}{4}$, and $\frac{d}{4} + \frac{1}{4}$. After the addition of one neuron ($k = d + 1$), there are 4 Hessian eigenvalues associated to the standard representation of $S_d$ and, modulo $O(d^{-\frac{1}{2}})$ terms, these satisfy

$$\frac{d}{4} + \frac{1}{2}, \frac{1}{4} - \frac{1}{2\pi}, \frac{-1 + \sqrt{5}}{4\pi} + \frac{1}{4} > 0 > \frac{-1 - \sqrt{5}}{4\pi} + \frac{1}{4}.$$

This example highlights the special role that the standard representation plays in the annihilation of spurious minima (see Section 5 and the concluding remarks). The sharp estimates of the Hessian spectrum further demonstrate how symmetry breaking enables a complete characterization of the dynamics of gradient-based methods, locally, in the vicinity of symmetric critical points. The dependence of such methods on stability of critical points therefore indicates that attempts for a global theory should be preceded by a good description of the mechanism by which spurious minima transform into saddles—the aim of this work.

Next, we relate our results to the existing literature.

**Annihilation of spurious minima on account of over-parameterization.** Existing methods for the analysis of optimization problem (2) include: mean-field [4], optimal transport [2], NTK [20, 21, 22] and the thermodynamic limit [5, 16, 23, 24]. These methods operate by passing to limiting regimes where the number of inputs or neurons is taken to infinity. A growing number of works has limited the explanatory power of such approaches [25, 26]. Approaches for addressing the loss landscapes in finite parameter regimes exist and include [6] which obtains several generalities on critical points, and [7] which studies conditions under which a single neuron can be added in contrived way so as to

turn a given minimum in $M(d, d)$ into a saddle in $M(d + 1, d)$. However, as shown in the present work, families of minima in $M(d + 1, d)$ exist. Indeed, families of minima of lesser symmetry are shown to require at least two additional neurons before turning into saddles.

**Symmetry breaking in nonconvex loss landscapes.** It has been recently found that the symmetry of spurious minima in optimization problem (2) *break the symmetry* of global minima [27] (see Section 2 for a formal exposition). A similar phenomenon has been observed for tensor decomposition problems in [28] and later studied [29]. The present work builds on methods developed in a line of work concerning this phenomenon of symmetry breaking. In [30], path based techniques are introduced to construct infinite families of critical points represented by Puiseux series in $d^{-1}$. In [31], results from the representation theory of the symmetric group are used, together with the Puiseux series result, to obtain precise analytic estimates on the Hessian spectrum. In [32], it is shown that certain families of saddles transform into spurious minima at a fractional dimensionality. Moreover, the spectra of these families of spurious minima is shown to be identical to that of global minima to $O(d^{-\frac{1}{2}})$-order. In [33], generic $S_d$-equivariant steady-state bifurcation is studied, emphasizing the complex geometry of the exterior square and the standard representations along which the minima studied in this work are created and annihilated.

The results above assume that the number of neurons is less or equal the number of inputs: $k \leq d$. The present work concerns the over-parameterized case $k > d$ (additional terms used in related contexts are *over-specified*, e.g., [34] and *over-realized*, e.g., [6]). The study of the technically more demanding case of over-parameterization requires new methods and ideas, which we describe below.

- We develop a method which allows the expression of eigenvalues in terms of the gradient entries. The approach is used to evaluate eigenvalues of $\Theta(d^2)$-multiplicity analytically, and reveals hidden algebraic relations between *criticality* and *curvature*.
- Eigenvalues of $O(d)$-multiplicity are computed numerically using estimates of the Puiseux series coefficients. The estimates are obtained through numerical methods used either directly for a system of equations corresponding to different orders of the Puiseux series terms, or for a reduced system of equations obtained by exploiting the geometric structure of the problem.
- Lastly, Cauchy's interlacing theorem is used to reduce the complexity of the computation of descent directions afforded by over-parameterization, and yields a tight characterization of linear subspaces along which spurious minima transform into saddles.

The methods are illustrated for eight families of critical points. Orthogonality of the target matrices is not required by the symmetry-breaking framework; other choices of target matrices, distributions, activation functions and architectures have been considered in previous works, and are a topic of current research.

**Organization of the paper.** The proof of Theorem 1 is in four parts: symmetry & the loss function, families of symmetric minima, Hessian spectrum, and a change of stability under over-parameterization. Proofs and technical details are deferred to the appendix.

## 2 Symmetry and the loss function

The presentation of our results requires some familiarity with group and representation theory. Key ideas and concepts are introduced as needed.

The *symmetric group* $S_d$, $d \in \mathbb{N}$, is the group of permutations of $[d] \doteq \{1, \ldots, d\}$. The *orthogonal group* $O(d)$ is the subgroup of all orthogonal linear maps on $\mathbb{R}^d$. We may identify $S_d$ with the subgroup of $O(d)$ consisting of permutation matrices. Thus $S_d$ acts naturally on $[d]$ (as permutations) and orthogonally on $\mathbb{R}^d$ (as permutation matrices).

For $k, d \in \mathbb{N}$, there are natural actions of $S_k$ and $S_d$ on the space $M(k, d)$ of $k \times d$ matrices: $S_k$ permutes rows, $S_d$ permutes columns. The loss function (2) is invariant under row permutations: $\mathcal{L}(\sigma W) = \mathcal{L}(W)$, for all $\sigma \in S_k$—whatever the choice of $V, \beta$. The product action of $S_k \times S_d$ on $M(k, d)$ plays a central role in the study of invariance properties of $\mathcal{L}$. If $A = [A_{ij}] \in M(k, d)$, $(\pi, \rho) \in S_k \times S_d$, then

$$(\pi, \rho)[A_{ij}] = [A_{\pi^{-1}(i), \rho^{-1}(j)}], \ \pi \in S_k, \ \rho \in S_d, \ (i, j) \in [k] \times [d]. \tag{3}$$

The action can be defined in terms of permutation matrices. If $k \geq d$, the *diagonal subgroup* $\Delta S_d$ of $S_k \times S_d$ is defined by $\Delta S_d = \{(g,g) \mid g \in S_d \subseteq S_k\}$. Clearly, $\Delta S_d \approx S_d$.

Henceforth assume $k \geq d$. Regard $M(d,d)$ as the linear subspace of $M(k,d)$ defined by appending $k - d$ zero rows to each matrix in $M(d,d)$. Let $V \in M(k,d)$ denote the matrix determined by the identity matrix $I_d \in M(d,d)$. By the orthogonal symmetry of the Gaussian distribution, our results will hold for any $V$ determined by a matrix in $\mathrm{O}(d)$. We shall not consider training processes but rather concrete instances of families of critical points, and so throughout weights are assumed fixed. The weights of the second layer of critical points studied in the present work consists of positive weights and so, by the positive homogeneity of the ReLU activation, there is no loss of generality in assuming that the second layer of weights is set to ones, i.e., $\boldsymbol{\alpha} = \boldsymbol{\beta} = \mathcal{I}_{k,1}$, $\mathcal{I}_{i,j}$ being the $i \times j$-matrix with all entries equal to 1.

Optimization problem (2) has a rich symmetry structure; for our choice of $V$, $\boldsymbol{\alpha}$ and $\boldsymbol{\beta}$, $\mathcal{L}$ is $S_k \times S_d$-invariant [27]. It is natural to ask how the critical points of $\mathcal{L}$ reflect this symmetry. Given $W \in M(k,d)$, the largest subgroup of $S_k \times S_d$ fixing $W$ is called the *isotropy* subgroup of $W$. The isotropy group quantifies the *symmetry* of $W$. If $k = d$, $V$ has isotropy group $\Delta S_d$ and every global minimizer of $\mathcal{L}$ lies on the $S_d \times S_d$-orbit of $V$ [7, 30]. Empirically, if $k \geq d$, non-degenerate (i.e., no zero Hessian eigenvalues) spurious minima of $\mathcal{L}$ tend to be highly symmetric in that their isotropy groups are conjugate to large subgroups of $\Delta S_d$. Indeed, we suspect that for our choice of $V$, the isotropy of non-degenerate spurious minima is always non-trivial and conjugate to a subgroup of $\Delta S_d$.

If $k > d$, the isotropy of $V$ is *not* a subgroup of $\Delta S_d$— it contains the subgroup $I_d \times S_{k-d}$ of row permutations. Perhaps surprisingly, $\mathcal{L}$ is more regular at critical points of spurious minima than at points giving the global minimum: if $k = d$, analyticity of $\mathcal{L}$ at $W = V$ fails. If $k > d$, $V$ has a row of zeros and $\mathcal{L}$ is not differentiable at $W = V$ [1] (see also [7]). It may easily be shown that there is a $k - d$-dimensional compact connected $S_k \times S_d$-invariant simplicial complex $\Lambda \subset M(k,d)$ consisting of all matrices that define the global minimum value of zero (see Section A.7). Necessarily, $\Lambda$ contains the $S_k \times S_d$ orbit of $V$. At boundary points of $\Lambda$, $\mathcal{L}$ is not differentiable. The Hessian, defined on the interior of the simplex, is *always* singular. Here our focus will always be on families of spurious minima with non-degenerate critical points.

# 3 Families of minima: structure and basic properties

Families of spurious minima often have characteristic properties. For example, the asymptotics in $d$ of the loss or their chance of being detected by SGD. For a systematic study of the optimization landscape of $\mathcal{L}$, we need to categorize minima and understand their distinctive analytic properties. Both isotropy and the notion of a *regular family* play an important role. Throughout we assume $k = d + m$, for all $d \geq d_0$, where $m \geq 0$ is an integer constant.

If $G$ is a subgroup of $\Delta S_d$, let $M(k,d)^G = \{W \in M(k,d) \mid gW = W, \forall g \in G\}$ denote the *fixed point space* for the action of $G$ on $M(k,d)$. Every $S_d \times S_k$-equivariant vector field on $M(k,d)$ is tangent to $M(k,d)^G$ and so $\nabla \mathcal{L}$ is tangent to $M(k,d)^G$. Hence $\mathfrak{c} \in M(k,d)^G$ is a critical point of $\mathcal{L}|M(k,d)^G$ iff $\mathfrak{c}$ is a critical point of $\mathcal{L}$. The inclusion $i_d : [d] \rightarrow [d+1]$ induces a natural inclusion $i_d : S_d \subset S_{d+1}$, where $i_d(S_d)$ fixes $d + 1$. More generally, given a positive integer $p$ and $d_0 > p$, we have a sequence of inclusions $i_{d,p} : S_{d-p} \times S_p \rightarrow S_{d+1-p} \times S_p$, $d \geq d_0$. Identifying $S_d$ with $\Delta S_d$, a sequence $(G_d)_{d \geq d_0}$ of subgroups of $\Delta S_d$ is *natural* if for some positive integer $p < d_0$, (a) $i_{d,p}(G_d) \subset G_{d+1}$, and (b) $\dim(M(k,d)^{G_d})$ is independent of $d \geq d_0$ (assume $k \geq d$).

**Example 1** *Set $G_d = \Delta(S_{d-p} \times S_p)$ and $m$ be a positive integer. If $p = 0$, then $G_d = \Delta S_d$ and $dim(M(d+m,d)^{G_d}) = 2+m$, $d \geq d_0 = 2$; if $p = 1$, then $dim(M(d+m,d)^{\Delta(S_{d-1} \times S_1)}) = 5+2m$, $d \geq 3$; if $p \geq 2$, then $dim(M(d+m,d)^{\Delta(S_{d-p} \times S_p)}) = 6+2m$, $d \geq 4$.*

If $(G_d)_{d \geq d_0}$ is natural, we often identify $M(k,d)^{G_d}$ with $\mathbb{R}^N$, $d \geq d_0$, where $\dim(M(k,d)^{G_d}) = N$. We define linear isomorphisms $\Xi : \mathbb{R}^N \rightarrow M(k,d)^{G_d}$ for the families of Example 1. Let $I_i^\star = \mathcal{I}_{i,i} - I_i$, $i \in \mathbb{N}$. The matrix $\Xi(\boldsymbol{\xi})$, $\boldsymbol{\xi} \in \mathbb{R}^N$, is expressed as block diagonal matrix $[B_{ij}]$, where each $B_{i,j}$ is a linear combination in the coordinates of $\boldsymbol{\xi}$ of the matrices $\mathcal{I}_{i,j}$, $I_i$ and $I_i^\star$. For example, if $p = 1$, $m \geq 1$, then $N = 5 + 2m$ and $\Xi(\xi_1, \cdots, \xi_N)$ is the $(2 + m) \times 2$-block matrix

$[B_{ij}] \in M(d+m, d)^{\Delta(S_{d-1} \times S_1)}$ defined by

$$B_{11} = \xi_1 I_{d-1} + \xi_2 I_{d-1}^\star, \; B_{12} = \xi_3 \mathcal{I}_{d-1,1},$$
$$B_{i1} = \xi_{2i} \mathcal{I}_{1,d-1}, \; B_{i2} = \xi_{2i+1} \mathcal{I}_{1,1}, \; 2 \le i \le 2+m.$$

Similar expressions hold for the other families. In practice, we restrict the vector field $\nabla \mathcal{L}$ to $M(k,d)^{G_d}$ and then pull back this vector field using $\Xi$ to a vector field $F_d$ on $\mathbb{R}^N$. The Jacobian of $F_d$ is then equal to the Hessian of $\nabla(\mathcal{L}|M(k,d)^{G_d})$. Observe that $F_d$ does not depend on a choice of inner product on $\mathbb{R}^N$; indeed, we take the standard Euclidean inner product on $\mathbb{R}^N$ ($\Xi : \mathbb{R}^N \to M(k,d)^G$ is *not* an isometry). Since $F_d(\boldsymbol{\xi}) = \Xi^{-1} \nabla \mathcal{L}(\Xi(\boldsymbol{\xi}))$, we may read off the components of $F_d(\boldsymbol{\xi})$ directly from the corresponding matrix entries of $\nabla \mathcal{L}$. We find that $F_d$ is a continuous family of vector fields on $\mathbb{R}^N$ in the *real* parameter $d$. Obviously, no such statement can hold on $M(k,d)$ as the dimension of $M(k,d)$ depends on $d$. Moreover, the vector fields $F_d$ will be real analytic outside of a thin (semianalytic) subset of $\mathbb{R}^N$. Indeed, $F_d$ is subanalytic [35] but the real analyticity statement is easily proved directly and suffices for our applications. All of this allows us to "connect" the critical points $\mathfrak{c}(d) \in M(k,d)^{G_d}$, $d \ge d_0$, by curves in $\mathbb{R}^N$ and develop a path-based approach to our problem.

**Definition 1** *(Notation & Assumptions as above.) A family $\{\mathfrak{C}(d) = \Xi(\mathfrak{c}(d)) \mid d \ge d_0\}$ of critical points of $\mathcal{L}$ with isotropy $G_d \subset \Delta S_d$ is* weakly regular *if $(G_d)_{d \ge d_0}$ is natural and for $d \ge d_0$*

   *(a) There is a continuous curve $\gamma_d : [0,1] \to \mathbb{R}^N$ of critical points of $F_d$ joining $\mathfrak{c}(d)$ to $\mathfrak{c}(d+1)$.*

   *(b) $\mathcal{L}$ is real analytic at $\Xi(\mathfrak{c}(d)) \in M(k,d)$.*

   *(c) $F_d$ is real analytic on a neighbourhood of $\gamma_d([0,1]) \subset \mathbb{R}^N$.*

   *(d) The Jacobian of $F_d$ along $\gamma_d$ is non-singular.*

*If $\lim_{d \to \infty} \mathfrak{c}(d) \doteq \mathfrak{c}_\infty \in \mathbb{R}^N$ exists and is bounded (Euclidean norm on $\mathbb{R}^N$), the family is* regular.

Using (c,d), and the real analytic implicit function theorem, $\gamma_d$ is real analytic. The family defined by $\mathfrak{c}(d) = V$ is not weakly regular as (b) fails even if $k = d$.

It may be shown, using results on subanalytic sets and the Curve Selection Lemma [36], that every regular family has a fractional power series (FPS) representation. That is, with the notation and assumptions of Definition 1, there exist $d_1 \ge d_0$ and a minimal $b \in \mathbb{N}$ such that each component $\mathfrak{c}_i(d)$ of $\mathfrak{c}(d) \in \mathbb{R}^N$ is given by the convergent power series

$$\mathfrak{c}(d)_i = \sum_{j=0}^{\infty} c_{i,j} d^{-\frac{j}{b}}, \; i \in [N]. \tag{4}$$

Under the assumption of weak regularity, there may be Puiseux series representations [30, Exam. 4.13] and $\mathfrak{c}_\infty$ lies in the one point compactification of $\mathbb{R}^N$. In practice, rather than use the general result, we prove directly that a family has an FPS representation. Verifying regularity for sufficiently large $d_0$, is usually straightforward or trivial. We refer to Section A for examples of construction of FPS representations when $k > d$.

The FPS representation for families of critical points is important both theoretically, and computationally and yields Puiseux series representations of the objective value and Hessian spectrum. It was shown in [30, Section 8] that for $k = d$ several (regular) families of critical points with isotropy $G_d = \Delta(S_{d-p} \times S_p)$, $p \in \{0,1\}$, had FPS representations in $d^{-\frac{1}{2}}$ (so $b = 2$). Each coordinate of $\mathfrak{c}_\infty = \lim_{d \to \infty} \mathfrak{c}(d) \in \mathbb{R}^N$ was either $\pm 1$ or zero. For examples of FPS representations with $k = d$ and $G_d = \Delta(S_{k-p} \times S_p)$, $p \in \{2,3\}$, see [32]. Explicit construction of the coefficients in these FPS examples is relatively straightforward and algebraic formulae can be given for low order terms. When $k > d$, analysis is harder. It is not always possible to give low order coefficients in a simple algebraic form. Moreover, there may be multiple regular families with the same limiting value $\mathfrak{c}_\infty \in \mathbb{R}^N$.

**Definition 2** *Let $p \ge 0$ and take $G_d = \Delta(S_{d-p} \times S_p)$ as in Example 1. A regular family of critical points with isotropy $(G_d)_{d \ge d_0}$ is of* type I *(resp.* type II*) if as $d \to \infty$, the diagonal elements of the $(d-p) \times (d-p)$-block corresponding to the action of $\Delta S_{d-p}$ converge to $-1$ (resp. $+1$).*

In terms of FPS, a family is of type I (resp. type II) if $c_{1,0} = -1$ (resp. $+1$).

**Remark 1** *When $k = d$, we suspect that the initial coefficients $c_{i,0}$ of the FPS of a regular family with isotropy $\Delta(S_{d-p} \times S_p)$, $p \geq 0$, satisfy $c_{i,0} \in \{\pm 1, 0\}$. However, this is false if $k > d$—we give an example later. Moreover if $k = d$, and with a slight extension of the notion of regular family, there is a regular family with $G_{2d} = \Delta(S_d \times S_d)$, $d \in \mathbb{N}$, with $c_{1,0} = +1$, $c_{5,0} = -1$ (coefficients corresponding to the diagonal entries of the principal blocks associated to $S_d \times I_d$ and $I_d \times S_d$).*

Henceforth, we emphasize $k > d$, and families with isotropy $\Delta(S_{d-p} \times S_p)$, $p \in \{0, 1\}$. However, the methods are quite general.

**Theorem 2** *Suppose that $\mathfrak{C}$ is a regular family of critical points. Assume that initial terms of the asociated FPS do not all vanish and $b \in \{2, 4\}$. If the isotropy $G_d = \Delta S_d$, $k \in \{d, d + 1\}$, then $\mathfrak{C}$ is of type I; if the isotropy $G_d = \Delta(S_{d-1} \times S_1)$, $k \in \{d, d + 1, d + 2\}$, then $\mathfrak{C}$ is either of type I or type II and there exists at least one family of each type. If $k = d$ there is precisely one type I family and, if $p \neq 0$, one type II family.*

**Remark 2** *For $k = d$, both type I and type II families are spurious minima [31, 32]. However, empirically, type I minima are not detected by SGD when Xavier initialization is used. Since the loss at type II minima decays as $\Theta(1/d)$ and the loss at type I is $\Theta(1)$ (independently of the isotropy), it may be tempting to argue that the expected initial loss under Xavier initialization is smaller than the loss at type I minima. However, this turns out to be false: Assume $k = d$. Under Xavier initialization, $\left(1 - \frac{2}{\pi}\right)d \leq E_W[\mathcal{L}(W)] \leq \left(1 - \frac{1}{\pi}\right)d$ (see Section A.6).*

As we increase $k - d$, the original type I and II critical points of spurious minima persist as degenerate critical sets and new regular families of critical points of the same type are generated. Thus, if $k - d = 1$, and $G_d \subsetneq \Delta S_d$, two regular families of type I points are generated which are swapped by the permutation of rows $d$ and $d + 1$. Similarly for families of type II. When $k - d = 2$, $3! = |S_3|$ new regular families of type I critical points appear; similarly for type II. Additional families of critical points, which do not originate from the original families and are not spurious minima, may appear. See Section A.7 for degenerate critical point sets occurring on account of over-parameterization.

Our focus will be on the families of type I and II critical points that arise through the above mechanism. For a given isotropy $G_d = \Delta(S_{d-p} \times S_p)$, $p \in \{0, 1\}$, a type X and $k \geq d$, with $m = k - d$ fixed, let $\mathfrak{C}_{p,m}^{X}$ denote a choice of regular family of critical points $\{\mathfrak{c}_{p,m}^{X}(d) \in M(k, d)^{G_p}\}$ that originates from the unique regular family of critical points of type X that exists when $k = d$. It is enough to analyze just one of the type X families when $k > d$ as, by equivariance, the choices lie on the same $S_d \times S_k$-orbit and so have similar Hessians. Once the existence of the FPS representation for the families $\mathfrak{C}_{p,m}^{X}$ has been proved, the next step is to estimate the Hessian spectrum, the topic of the next section.

We conclude with examples illustrating the quantitative power of our approach.

**Example 2** *(1) We investigate how the loss $\mathcal{L}(\mathfrak{c}_{1,m}^{II})$ depends on $k - d$ for the type II families $\mathfrak{C}_{1,m}^{II}$, $m \in \{0, 1, 2\}$. For $m > 0$, the initial coefficients of the FPS are found using Newton-Raphson method applied either directly for a system of equations corresponding to different orders of the FPS coefficients Section A.5, or for a reduced system of equations obtained through an explicit use of the geometry of the problem Section A.1. We give the asymptotics modulo $O(d^{-\frac{3}{2}})$ and find that if $\mathcal{L}(\mathfrak{c}_{1,m}^{II}) = \alpha_m d^{-1} + O(d^{-\frac{3}{2}})$, then*

$$\alpha_0 = 2.97357632715\ldots \left(= \frac{1}{2} - \frac{2}{\pi^2}\right), \ \alpha_1 = 2.67254813889\ldots, \ \alpha_2 = 2.67193392202\ldots$$

*(2) Consider $\mathcal{L}(\mathfrak{c}_{1,m}^{I})$ for type I families. We find that $\mathcal{L}(\mathfrak{c}_{1,m}^{I}) = \frac{1}{2} - \frac{1}{\pi} + O(d^{-\frac{1}{2}})$, for $m \in \{0, 1, 2\}$. Higher order terms are $m$-dependent but can be computed, as they can if $p = 0$. For example, if $p = 0$, $m = 1$ then $\mathcal{L}(\mathfrak{c}_{0,1}^{I}) = \frac{1}{2} - \frac{1}{\pi} - \frac{4}{3\pi}d^{-\frac{1}{2}} + \left(-1 - \frac{2}{\pi^2} + \frac{4}{\pi}\right)d^{-1} + O(d^{-\frac{3}{2}})$.*

*(3) We conclude with an example where the FPS is in powers of $d^{-\frac{1}{4}}$ and two components of $\mathfrak{c}_\infty \in \mathbb{R}^7$ do not lie in $\{\pm 1, 0\}$. If $k = d + 1$, the initial terms of the FPS for a type I family $\mathfrak{c}(d)$ of critical*

*points of spurious minima are given by*

$$\mathfrak{c}(d)_1 = -1 + 2d^{-1} + \frac{\pi}{2}d^{-\frac{3}{2}} + O(d^{-\frac{7}{4}}), \quad \mathfrak{c}(d)_2 = 2d^{-1} - \sqrt{\pi - 2}d^{-\frac{7}{4}} + O(d^{-2}),$$

$$\mathfrak{c}(d)_3 = d^{-1} - \frac{6 + 3\pi}{4\pi\sqrt{\pi - 2}}d^{-\frac{5}{4}} + O(d^{-\frac{3}{2}}), \quad \mathfrak{c}(d)_4 = \frac{\sqrt{\pi - 2}}{2}d^{-\frac{3}{4}} + O(d^{-1}),$$

$$\mathfrak{c}(d)_5 = \frac{1}{2} + \frac{6 + 3\pi}{8\pi\sqrt{\pi - 2}}d^{-\frac{1}{4}} + O(d^{-\frac{1}{2}}), \quad \mathfrak{c}(d)_6 = \frac{\sqrt{\pi - 2}}{2}d^{-\frac{3}{4}} + O(d^{-1}),$$

$$\mathfrak{c}(d)_7 = -\frac{1}{2} + \frac{6 + 3\pi}{8\pi\sqrt{\pi - 2}}d^{-\frac{1}{4}} + O(d^{-\frac{1}{2}})$$

*(see Section A.5 for details). The loss* $\mathcal{L}(\mathfrak{c}_{1,1}^I) = \frac{1}{2} - \frac{1}{\pi} - \frac{4}{3\pi}d^{-\frac{1}{2}} - \frac{(\pi-2)^{\frac{3}{2}}}{3\pi}d^{-\frac{3}{4}} + O(d^{-1}).$

## 4 Hessian spectrum

The FPS representation makes possible an analytic characterization of the Hessian spectrum using tools from the representation theory of groups (see Section B for a brief review). The main tool used is the *isotypic decomposition* relating the isotropy of a given point to minimal invariant subspaces of the Hessian. We begin by presenting the isotypic decomposition needed for the over-parameterized case.

Let $k \geq d$. Regard $M(k, d)$ as an $S_d$-representation (diagonal action on $M(d, d) \subset M(k, d)$). By restriction, $M(k, d)$ is a $S_q \times S_p$-representation, where $q = d - p$, $p < q$, and $S_q \times S_p \subset S_d$. If $p \in \{0, 1\}$ (the case of interest here) the isotypic decomposition uses 4 irreducible representations of $S_q$, when $d \geq 4$: the trivial representation $\mathfrak{t}$ of degree 1, the standard representation $\mathfrak{s}_q$ of $S_q$ of degree $q - 1$, the exterior square representation $\mathfrak{x}_q = \wedge^2\mathfrak{s}_q$ of degree $\frac{(q-1)(q-2)}{2}$ and a representation $\mathfrak{y}_q$ of degree $\frac{q(q-3)}{2}$ (associated to the partition $(q - 2, 2)$ [37, 38]). For $p \in \{0, 1\}$, the isotypic decomposition is

$$M(k, d) = (m + 3p + 2)\mathfrak{t} + (pm + 2p + 3)\mathfrak{s}_q + \mathfrak{x}_q + \mathfrak{y}_q. \tag{5}$$

Since the representations $\mathfrak{x}_q, \mathfrak{x}_q$ contribute 2 eigenvalues, of total multiplicity $q^2 - 2q + 1$, we have

**Lemma 1** *If $k - d = m \geq 0$ and $p \in \{0, 1\}$, then of the $kd$ eigenvalues of the Hessian at a point of isotropy $\Delta(S_{d-p} \times S_p)$:*

    *1. $\Theta(d^2)$ are populated by two eigenvalues: the $\mathfrak{x}$- and $\mathfrak{y}$-representation eigenvalues.*

    *2. At most $O(k - d)$ eigenvalues are distinct.*

Lemma 1 implies that the isotropy type of a point strictly restricts the number of distinct eigenvalues of the Hessian spectrum. For fixed $k - d$, the $\mathfrak{x}$- and the $\mathfrak{y}$-representation eigenvalues account for $kd - \Theta(d)$ of the eigenvalues. We show that for all families of critical points considered here, the $\mathfrak{x}$- and the $\mathfrak{y}$-representation eigenvalues are identical to order $O(d^{-\frac{1}{4}})$.

**Theorem 3** *For a family of critical points of isotropy $\Delta(S_{d-p} \times S_p)$, $p \in \{0, 1\}$ and $k$ as in Theorem 2, $kd - \Theta(d)$ of the Hessian eigenvalues are populated by the two eigenvalues:*

$$\frac{1}{4} - \frac{1}{2\pi} + O(d^{-\frac{1}{4}}) \quad and \quad \frac{1}{4} + \frac{1}{2\pi} + O(d^{-\frac{1}{4}})$$

*associated to the $\mathfrak{x}$- and the $\mathfrak{y}$-representation, respectively.*

The derivation of Theorem 3 builds on a technique used in [31, 32] and is directed towards the case where $k > d$ and the coefficients of FPS may not be given in a simple algebraic form. Specifically, we rewrite the expression for the Hessian eigenvalues in terms of the gradient entries. Since gradient entries vanish at critical points, this allow us to evaluate the eigenvalue expressions. For example, the Puiseux series of the $\mathfrak{x}$-eigenvalue of Type I $\Delta(S_{d-1} \times S_1)$-critical points is

$$\lambda_{\mathfrak{x}}^d = \frac{1}{4} - \frac{1}{2\pi} + [d^0]F_{d,1} - c_{1,2}[d^{\frac{1}{2}}]F_{d,1} - [d^0]F_{d,2} + [d^{\frac{1}{4}}]F_{d,1}d^{\frac{1}{4}} + [d^{\frac{1}{2}}]F_{d,1}d^{\frac{1}{2}} + O(d^{-\frac{1}{4}}), \tag{6}$$

with $[d^\alpha]F_{d,i}$ indicating the coefficient of $d^\alpha$ in $F_{d,i}$. Since $F_{d,i}$ vanish at critical points, $\lambda_{\mathfrak{r}}^d = \frac{1}{4} - \frac{1}{2\pi} + O(d^{-\frac{1}{4}})$. Equation 6 further demonstrates the sensitivity of the $\mathfrak{r}$-eigenvalue to variations in the FPS coefficients and in different orders of the gradient terms, see Section B.1. Algebraic relations between criticality and curvature indicate therefore a certain rigidity of the loss landscape. Relations of similar nature exist between *criticality* and the *loss* at a point.

It follows from Theorem 3 that all Hessian eigenvalues not associated to the trivial or standard representations are strictly positive for sufficiently large $d$. Consequently, annihilation of spurious minima in a family must be tangent to an invariant subspace of the sum of the isotypic components for the trivial and standard representations. Generically, it is to be expected that the subspace will be isomorphic to either the trivial representation or the standard representation.

## 5  Over-parameterization

Having computed the $\mathfrak{r}$- and the $\mathfrak{y}$-eigenvalues, we now turn to describe how the eigenvalues associated to the trivial and the standard representations vary when the number of neurons is increased. We find that while eigenvalues associated to the trivial representation remain strictly positive for all sufficiently large $d$—some eigenvalues, associated to the standard representation, become negative, indicating a transition from minima to saddles along the isotypic component of the standard representation. We start with points of isotropy $\Delta S_d$.

### 5.1  Critical points of isotropy $\Delta S_d$

By Theorem 2, if $k \in \{d, d+1\}$, there is one regular family of critical points with isotropy $\Delta S_d$: the type I family $\mathfrak{C}_{0,i}^{\mathrm{I}}$, $i = k - d$. The representation-theoretic tools used in Section 4, yield a complete characterization of the Hessian spectrum of both families of critical points (see the discussion following the statement of Theorem 1 in the introduction for more details).

The spectral analysis of the Hessian reveals that:

A. $\mathfrak{C}_{0,0}^{\mathrm{I}}$ is a family of minima.

B. Adding one neuron turns it into the family $\mathfrak{C}_{0,1}^{\mathrm{I}}$ of non-degenerate saddles where the negative eigenvalue of the Hessian at $\mathfrak{C}_{0,1}^{\mathrm{I}}$ is associated to the standard representation $\mathfrak{s}_d$.

Since the negative eigenvalue of the Hessian at $\mathfrak{C}_{0,1}^{\mathrm{I}}$ is associated to $\mathfrak{s}_d$, there are exactly $d-1$ descent directions, out of $d(d+1)$ possible directions in $M(d+1, d)$, lying in the $4d-4$-dimensional isotypic component $4\mathfrak{s}_d$ spanned by (only nonzero elements are described):

1. The $(d-1)$-dimensional space of $(d+1) \times d$-matrices $[y_{ij}]$ where for $i, j \in [d]$, $y_{ij} = z_i - z_j$, for some $(z_1, \cdots, z_d) \in \mathbb{R}^d$ with $\sum_{i \in [d]} z_i = 0$.

2. The $(d-1)$-dimensional space of $(d+1) \times d$-matrices $[y_{ij}]$ where for $i, j \in [d]$, $i \neq j$, $y_{ij} = z_i + z_j$, where $(z_1, \cdots, z_d) \in \mathbb{R}^d$ with $\sum_{i \in [d]} z_i = 0$.

3. The $(d-1)$-dimensional space of $(d+1) \times d$-matrices whose diagonal elements sum up to zero.

4. The $(d-1)$-dimensional space of $(d+1) \times d$-matrices whose $(d+1)$'th row elements sum up to zero.

No regular families with isotropy $\Delta S_d$ exists if two neurons or more are added, i.e., $k - d \geq 2$ (see Section A.7).

### 5.2  Critical points of isotropy $\Delta(S_{d-1} \times S_1)$

Consider the type II regular families $\mathfrak{C}_{1,0}^{\mathrm{II}}$, $\mathfrak{C}_{1,1}^{\mathrm{II}}$ and $\mathfrak{C}_{1,2}^{\mathrm{II}}$, corresponding to $k = d, d+1$ and $d+2$ respectively. We show that negative eigenvalues of the Hessian appear when $k - d = 2$ but not $k - d = 1$. The same result (and proof) hold for the type I family.

For $k = d, k = d+1$, the eigenvalues associated to the trivial and standard representation $\mathfrak{s}_{d-1}$ are strictly positive—see Table 5.2. By Theorem 3, the $\mathfrak{r}$- and the $\mathfrak{y}$-representation eigenvalues are also

| Isotypic component | Degree | $k = d$ | $k = d+1$ |
|---|---|---|---|
| | | Symmetry $\Delta S_{d-1}$ | Symmetry $\Delta S_{d-1}$ |
| Trivial representation | 1 | 0.0908 | 0.0044, 0.0843 |
| | | 0.25 | 0.2632, 0.3121 |
| mult. 5, if $k = d$ | | $0.1591d - 0.3471$ | $0.1591d + 0.7546$ |
| mult. 7, if $k = d+1$ | | $0.25d + 0.25$ | $0.25d + 0.5$ |
| | | $0.25d + 0.8471$ | $0.25d + 1.0979$ |
| Standard representation $\mathfrak{s}_{d-1}$ | $d-2$ | 0.0908 | 0.0230, 0.0908 |
| | | 0.0908 | 0.0936 |
| mult. 5, if $k = d$ | | 0.25 | 0.2693 |
| mult. 6, if $k = d+1$ | | 0.4091 | 0.5340 |
| | | $0.25d + 0.25$ | $0.25d + 0.5$ |
| Loss | | $\Theta(1/d)$ | $\Theta(1/d)$ |

Table 1: Type II critical points with symmetry $\Delta S_{d-1}$. The Hessian eigenvalues associated to the trivial and the standard representation are given to four decimal places, modulo $O(d^{-\frac{1}{2}})$-terms.

strictly positive. Therefore, we have families of spurious minima for $k = d, d+1$. For $k = d+2$, critical points in the family $\mathfrak{C}^{II}_{1,2}$ are saddles, with strictly negative eigenvalues associated to $\mathfrak{s}_{d-1}$. To show this, we take a different route so as to reduce the complexity of the computation. Consider the $2d \times 2d$-submatrix $\widehat{H}$ of the Hessian corresponding to the last two rows of the weight matrix. By Cauchy's interlacing theorem [39], the smallest eigenvalue of the Hessian is bounded from above by the smallest eigenvalue of $\widehat{H}$. Therefore, it suffices to prove that $\widehat{H}$ has a negative eigenvalue. Since the isotypic decomposition corresponding to $\widehat{H}$ consists of exactly two of the subspaces associated to $\mathfrak{s}_{d-1}$, the problem is reduced to computing the spectrum of a $2 \times 2$ matrix. Using Puiseux series representation, we show that modulo $O(d^{-\frac{1}{2}})$-terms the two eigenvalues of $\hat{H}$ are $\lambda_1 = 0.8060\ldots$ and $\lambda_2 = -0.1198\ldots$. Hence there exists a $(d-2)$-dimensional eigenspace of descent directions, projecting onto the associated eigenspace for $\widehat{H}$. Applying Cauchy's interlacing theorem again, there must also exist positive eigenvalues associated to $\mathfrak{s}_{d-1}$.

**Theorem 4** *(Notation & Assumptions as above.)*

- *$\Delta S_d$-symmetric critical points of type I are minima for $k = d$ and non-degenerate saddles for $k = d+1$ with a $(d-1)$-dimensional eigenspace of descent directions.*

- *Critical points of isotropy $\Delta(S_{d-1} \times S_1)$, type I or II, define regular families of spurious minima for $k = d, d+1$, and non-degenerate saddles for $k = d+2$, with at least a $(d-2)$-dimensional space of descent directions.*

Empirically, when $k = d+1$, minima of isotropy $\Delta(S_{d-1} \times S_1)$ are not seen for $d < 8$ (they are if $k = d$ and $d \geq 6$) and the probability to detect them using gradient descent is much lower for small values of $d \geq 8$ [19] (type I minima are not detected under Xavier initialization). Theorem 4 implies that when $k = d+2$ there are no spurious minima of type II of symmetry $\Delta(S_{d-1} \times S_1)$ and the descent directions are tangent to a copy of $\mathfrak{s}_{d-1}$. The last point is crucial for understanding the empirical results. The small eigenvalue 0.0230 associated to $\mathfrak{s}_{d-1}$ when $k = d+1$ indicates that we are close to a change of stability (bifurcation) of the critical point for gradient descent. Bifurcation of the trivial solution on $\mathfrak{s}_{d-1}$ is special and quite exceptional. In our case, the trivial solution will be a sink for gradient descent (i.e., a strict local minimum of the loss function), when $k = d+1$, and a source (i.e., a strict local maximum of the loss function) when $k = d+2$. The change of stability results from the collision of a large number of saddles of *high index* with the sink, followed by the emergence of a source and a large number of saddles with low index. The high index of the saddles converging to the sink, implies that the basin of attraction for the sink shrinks rapidly as the saddles approach the sink. We refer to [33, Sections 1.1, 4] for more on this phenomenon. As we increase $d$, families $\mathfrak{C}^{II}_p, p > 1$, of spurious minima appear which may not be annihilated by adding two neurons. However, no such minima were found in [19] when $k = d+2, d \leq 20$ (they were for $k = d+1$). The empirical results provide strong support for a change of stability associated to $\mathfrak{s}_{d-1}$ and suggest the unique role this representation may play in understanding over-parametrization.

## Concluding Comments

The rich symmetry structure exhibited by the loss function (2) makes possible an analytic study of the associated nonconvex loss landscape. The approach is twofold. First, the presence of symmetry breaking allows an efficient organization of an otherwise highly complex set of critical points, offering new ways of recognizing, differentiating and understanding local minima (Section 2 and Section 3). Second, for a given family of critical points, symmetry grants a parameterization of fixed dimensionality, independent of the ambient space, which permits a new array of analytic and algebraic tools (Section 4 and Section 5).

In this work, the symmetry breaking framework is used for investigating the nature by which over-parameterization contributes to making the loss landscape of (2) accessible for gradient-based methods. We find that increasing the number of neurons transforms spurious minima into saddles: decent directions are formed along linear subspaces corresponding the standard representation $\mathfrak{s}_d$ of $S_d$, ascent directions along other representations of $S_d$ persist, and the loss remains essentially the same. The process by which spurious minima turn into saddles suggests a powerful mechanism enabling minimization of the *loss* (rather than the gradient norm [40, 41, 42]) via computationally efficient local search methods, and further highlights the importance of the intricate interplay between symmetries inherent to data distributions and those displayed by neural network models.

Our spectral results assume the target $V$ has high symmetry but apply also to asymmetric problems. In the cases we discuss, the transition from saddle to minimum, or minimum to saddle, occurs at a non-integer value of $d$. Hence, at integer values of $d$, critical points are non-degenerate. It follows that for the families $\mathfrak{C}_{p,m}^X$ we consider, there is an open neighborhood $\mathcal{U}$ of $V = I_d \in M(d,d)$, such that for all $V' \in \mathcal{U}$, the loss function $\mathcal{L}'$ for $V'$ has a non-degenerate critical point close to each point of the $\Delta(S_{d-p} \times S_p)$-orbit of $\mathfrak{C}_{p,m}^X$ with the same number of negative eigenvalues (counting multiplicities). Critical points and eigenvalues depend continuously on $V' \in \mathcal{U}$ and the Hessian spectrum remains extremely skewed.

There is the problem of understanding the geometric mechanisms underlying the transition from minimum to saddle. As already indicated, this is closely related to the geometry of the standard representation $\mathfrak{s}_d$ of $S_d$. For simplicity, assume $d = 2\ell + 1$ is odd (similar results hold if $d$ is even [33]). Gradient vector fields on $\mathfrak{s}_d$ always have a critical point at the origin which is never a non-degenerate saddle but is (generically) either a non-degenerate minimum or maximum. The transition between minima (sink for the gradient descent, index $2\ell$) and maxima (source, index 0) can be achieved *locally* (that is, as a local deformation of the landscape geometry) by $2^{d-1} - 1$ non-degenerate saddles of index $\geq \ell$ passing simultaneously through the origin and emerging as $2^{d-1} - 1$ non-degenerate saddles of index $\leq \ell$. *No new minima or maxima are created.* Forced symmetry breaking leads to great complexity near the transition but minimal models of complexity can be given (*op. cit.*, Section 4).

Rather than striving for generalization, our approach in this work has been to focus on an analytically tractable case, one already acknowledged as difficult [2, 4, 5, 9, 19, 20], that helps elucidate some of the key foundational questions. The phenomena described are robust and so already have the power to disprove or support general conjectures in DL [30, 31, 32]. The symmetry breaking framework used to study these phenomena generalizes beyond the families of minima considered in the present work [43], and applies to other choices of activation functions and distributions [27, 28]. In addition, numerical work indicates that minima of the empirical loss are also symmetry breaking, and so allow theoretical investigations of the empirical loss landscape as well as algorithmic biases (see, e.g., [44]) within the new analytic framework. The full scope and power of symmetry breaking in DL, and more generally stochastic nonconvex optimization, remain to be discovered.

## Acknowledgments and Disclosure of Funding

We thank Noa Aharon, Daniel Soudry and Avi Wigderson for valuable discussions and constructive suggestions. The research was supported by the Israel Science Foundation (grant No. 724/22).

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
