# A  Supplementary material for Section 3

We begin by outlining the proof of existence for type II families with $k = d + 2$ and isotropy $\Delta(S_{d-1} \times S_1)$ following the strategy in [30, Section 8], and briefly discuss other cases given in Theorem 2. Estimates required for a valid use of the real analytic version of the implicit function theorem are obtained using two different methods presented in Section A.1 and in Section A.5.

## A.1  Fractional power series representation for type II minima: $k = d + 2$

Following Section 3, we restrict the loss $\mathcal{L}$ to the fixed point space $M(d + 2, d)^{\Delta S_{d-1}}$, which is of dimension 9 for $d \geq 3$. We recall the linear isomorphism $\Xi : \mathbb{R}^9 \to M(d + 2, d)^{\Delta S_{d-1}}$ defined by

$$\Xi(\boldsymbol{\xi}) = \begin{bmatrix} \xi_1 \mathcal{I}_{d-1,d-1} + \xi_2 I_{d-1}^\star & \xi_3 \mathcal{I}_{d-1,1} \\ \xi_4 \mathcal{I}_{1,d-1} & \xi_5 \mathcal{I}_{1,1} \\ \xi_6 \mathcal{I}_{1,d-1} & \xi_7 \mathcal{I}_{1,1} \\ \xi_8 \mathcal{I}_{1,d-1} & \xi_9 \mathcal{I}_{1,1} \end{bmatrix}, \quad \boldsymbol{\xi} = (\xi_1, \cdots, \xi_9) \in \mathbb{R}^9$$

Regard $\Xi$ as an identification of $M(d + 2, d)^{\Delta S_{d-1}}$ with $\mathbb{R}^9$ and recall from Section 3 that $F_d$ is the vector field on $\mathbb{R}^9$ defined as the pullback of $\nabla \mathcal{L}|M(k, d)^{\Delta S_{d-1}}$ by $\Xi$.

The map $\Xi$ naturally determines a $4 \times 2$-block decomposition of matrices in $M(d + 2, d)^{\Delta S_{d-1}}$. If a row $\boldsymbol{w}$ of $W \in M(d + 2, d)^{\Delta S_{d-1}}$ lies in row $\ell$ of the block decomposition, the row is said to be of *row-type* $\ell$. Necessarily $\ell \in [4]$. Clearly there are $d - 1$-rows of row-type 1 and exactly one row of row-type $\ell$, $2 \leq \ell \leq 4$.

**Remark 3** *The vector fields $(\nabla \mathcal{L})|M(k, d)^G$, $\nabla(\mathcal{L}|M(k, d)^G)$ on $M(k, d)^G$ are equal,* provided that we use the inner product on $M(k, d)^G$ induced from $M(k, d)$ to define the gradient of $\mathcal{L}|M(k, d)^G$, and so the eigenvalues of the Hessian of $\mathcal{L}$ at $\mathfrak{c}$ corresponding to directions tangent to $M(k, d)^G$ will equal the eigenvalues of the Hessian of $\mathcal{L}|M(k, d)^G$ at $\mathfrak{c}$. Indeed, the Jacobian of $(\nabla \mathcal{L})|M(k, d)^G$ (or $F_d$) is equal to the Hessian of $\mathcal{L}|M(k, d)^G$.

Let $S_3$ denote the subgroup of $S_k$ permuting the last three rows of $M(d + 2, d)$. Suppose $\mathfrak{c}$ is a type II critical point of $\mathcal{L}$. Since $M(d + 2, d)^{\Delta S_{d-1}}$ is invariant, but not fixed, by the $S_3$-action, it follows by the $S_k \times S_d$-invariance of $\mathcal{L}$ that the $S_3$-orbit of $\mathfrak{c}$ is a subset of $M(d + 2, d)^{\Delta S_{d-1}}$ containing at most six points ($|S_3| = 6$).

We claim that if $\mathfrak{c}$ is fixed by a non-identity element of $S_3$, then the Hessian of $\mathfrak{c}$ has to be singular. This follows since otherwise at least two of the final three rows of $\mathfrak{c}$ must be parallel and so $\mathfrak{c}$ lies in a set of critical points of $\mathcal{L}|M(k, d)^{\Delta S_{d-1}}$ of dimension at least one (see Section A.7 below). In particular, for a regular family, (a) critical points cannot be fixed by a non-identity element of $S_3$, and (b) if $\mathfrak{C}$ is a regular family in $M(d + 2, d)^{\Delta S_{d-1}}$, so is $\sigma \mathfrak{C}$, $\sigma \in S_3$, and their paths do not cross. Generally, we regard the six type II families that result from this observation as being essentially the same and focus on just one of them. That choice comes naturally from an order on the critical points that we discuss shortly. Unlike type I families, the type II family has a rich geometric structure that plays a significant role in the analysis.

The next step is to give an explicit expression for the gradient vector field of $\mathcal{L}$ restricted to $M(d + 2, d)^{\Delta S_{d-1}} \approx \mathbb{R}^9$. Given $W \in M(d + 2, d)^{\Delta S_{d-1}}$, denote the rows of $W$ by $\boldsymbol{w}_1, \cdots, \boldsymbol{w}_{d+2}$. After defining norms, dependent only the row-type, we define angles between different rows of $W$ (capital Greek) and angles between rows of $W$ and rows of $V$ (lower case Greek). Two subscripts are needed for angles between rows of different row-type, one subscript suffices for the same row type. For $p \in [3]$, it is often convenient to set $d_p = d - p$.

For $a \in [4]$, let $\tau_a$ denote the norm of $\boldsymbol{w}_i \in M(d + 2, d)^{\Delta S_{d-1}}$, where $\boldsymbol{w}_i$ is of row-type $a$ and the norm induced from standard Euclidean inner product on $M(d + 2, d)$. In $\mathbb{R}^N$ coordinates, $\tau_1 = \sqrt{\xi_1^2 + d_2 \xi_2^2 + \xi_3^2}$, $\tau_2 = \sqrt{d_1 \xi_4^2 + \xi_5^2}$, etc.

For $i, j \in [d - 1]$, $i \neq j$, let $\Theta_1$ denote the angle between rows $\boldsymbol{w}_i$ and $\boldsymbol{w}_j$ and $\theta_1$ denote the angle between row $\boldsymbol{w}_i$ and $\boldsymbol{v}_j$ (here and below, angles are well-defined independently of $i, j$ using symmetry). For $a, b \in [4]$, $a \neq b$, denote the angle between $\boldsymbol{w}_a$ and $\boldsymbol{w}_b$ by $\Lambda_{ab}$ (that is, the angle between rows of row-types $a$ and $b$, $a \neq b$). For $a \in [4]$, $b \in [2]$, let $\lambda_{ab}$ denote the angle between a $W$-row of type $a$ and a $V$-row of type $b$, where if $a = b \leq 2$, we assume $W, V$ are different rows.

Note that $\Lambda_{ab}$ is symmetric in $a, b$ but $\lambda_{ab}$ is not; indeed, $\lambda_{ba}$ is not even defined if $a > 2$. Finally, let $\beta_1$ denote the angle between $\boldsymbol{w}_i$ and $\boldsymbol{v}_i$, where $i \in [d-1]$, and $\beta_2$ denote the angle between $\boldsymbol{w}_d$ and $\boldsymbol{v}_d$. All these angles may be expressed as inverse cosines of expressions in the variable $\xi_1, \cdots, \xi_9$. For example

$$\Lambda_{24} \doteq \cos^{-1}\left(\frac{\langle w^d, w^{d+2}\rangle}{\tau_2 \tau_4}\right) = \cos^{-1}\left(\frac{d_1 \xi_4 \xi_8 + \xi_5 \xi_9}{\tau_2 \tau_4}\right)$$

The norms and angles are well defined—depend only row-type—on account of symmetry.

Next we give expressions for the vector field induced on $\mathbb{R}^9$ by $\nabla\mathcal{L}|M(d+2,d)^{\Delta S_{d-1}}$. Define

$$\Gamma_1(\boldsymbol{\xi}) = d_2\left(\frac{\tau_1 \sin(\Theta_1) - \sin(\theta_1)}{\tau_1}\right) + \frac{[\sum_{j \neq 4}\tau_j \sin(\Lambda_{1j})] - \sin(\lambda_{12}) - \sin(\beta_1)}{\tau_1}$$

$$\Gamma_2(\boldsymbol{\xi}) = d_1\left(\frac{\tau_1 \sin(\Lambda_{21}) - \sin(\lambda_{21})}{\tau_2}\right) + \frac{\tau_3 \sin(\Lambda_{23}) + \tau_4 \sin(\Lambda_{24}) - \sin(\beta_2)}{\tau_2}$$

$$\Gamma_3(\boldsymbol{\xi}) = d_1\left(\frac{\tau_1 \sin(\Lambda_{31}) - \sin(\lambda_{31})}{\tau_3}\right) + \frac{\tau_2 \sin(\Lambda_{32}) + \tau_4 \sin(\Lambda_{34}) - \sin(\lambda_{32})}{\tau_3}$$

$$\Gamma_4(\boldsymbol{\xi}) = d_1\left(\frac{\tau_1 \sin(\Lambda_{41}) - \sin(\lambda_{41})}{\tau_4}\right) + \frac{\tau_2 \sin(\Lambda_{42}) + \tau_3 \sin(\Lambda_{43}) - \sin(\lambda_{42})}{\tau_4}$$

Define nine "angle" terms.

$A_1^1 = d_2\Theta_1\xi_2 + \Lambda_{i2}\xi_4 + \Lambda_{13}\xi_6 + \Lambda_{14}\xi_8 - \beta_1$, $A_\star^1 = (\xi_1 + d_3\xi_2)\Theta_1 + \Lambda_{12}\xi_4 + \Lambda_{13}\xi_6 + \Lambda_{14}\xi_8 - \theta_1$

$A_2^1 = d_2\Theta_1\xi_3 + \Lambda_{12}\xi_5 + \Lambda_{13}\xi_7 + \Lambda_{14}\xi_9 - \lambda_{12}$

$A_1^2 = (\xi_1 + d_2\xi_2)\Lambda_{12} + \Lambda_{23}\xi_6 + \Lambda_{24}\xi_8 - \lambda_{21}$, $A_2^2 = d_1\Lambda_{12}\xi_3 + \Lambda_{23}\xi_7 + \Lambda_{24}\xi_9 - \beta_2$

$A_1^3 = (\xi_1 + d_2\xi_2)\Lambda_{13} + \Lambda_{23}\xi_4 + \Lambda_{34}\xi_8 - \lambda_{31}$, $A_2^3 = d_1\Lambda_{13}\xi_3 + \Lambda_{23}\xi_5 + \Lambda_{34}\xi_9 - \lambda_{32}$

$A_1^4 = (\xi_1 + d_2\xi_2)\Lambda_{14} + \Lambda_{24}\xi_4 + \Lambda_{34}\xi_6 - \lambda_{41}$, $A_2^4 = d_1\Lambda_{14}\xi_3 + \Lambda_{24}\xi_5 + \Lambda_{34}\xi_7 - \lambda_{42}$

Finally, define

$$\Omega_1 = \pi[\xi_1 + d_2\xi_2 + \xi_4 + \xi_6 + \xi_8 - 1]$$
$$\Omega_2 = \pi[d_1\xi_3 + \xi_5 + \xi_7 + \xi_9 - 1]$$

Note that $\Omega_1$ is the column sum of any one of the first $d-1$ columns of the matrix $\pi(\Xi(\boldsymbol{\xi}) - V)$ and $\Omega_2$ is the sum of column $d$ of $\pi(\Xi(\boldsymbol{\xi}) - V)$.

The components $(F_{d,1}, \cdots, F_{d,9})$ of $F_d$ are given by

$$F_{d,1}(\boldsymbol{\xi}) = \frac{1}{2\pi}(\Gamma_1\xi_1 - A_1^1 + \Omega_1) \qquad F_{d,2}(\boldsymbol{\xi}) = \frac{1}{2\pi}(\Gamma_1\xi_2 - A_\star^1 + \Omega_1)$$

$$F_{d,3}(\boldsymbol{\xi}) = \frac{1}{2\pi}(\Gamma_1\xi_3 - A_2^1 + \Omega_2)$$

$$F_{d,4}(\boldsymbol{\xi}) = \frac{1}{2\pi}(\Gamma_2\xi_4 - A_2^1 + \Omega_1) \qquad F_{d,5}(\boldsymbol{\xi}) = \frac{1}{2\pi}(\Gamma_2\xi_5 - A_2^2 + \Omega_2)$$

$$F_{d,6}(\boldsymbol{\xi}) = \frac{1}{2\pi}(\Gamma_3\xi_6 - A_1^3 + \Omega_1) \qquad F_{d,7}(\boldsymbol{\xi}) = \frac{1}{2\pi}(\Gamma_3\xi_7 - A_2^3 + \Omega_2)$$

$$F_{d,8}(\boldsymbol{\xi}) = \frac{1}{2\pi}(\Gamma_4\xi_8 - A_1^4 + \Omega_1) \qquad F_{d,9}(\boldsymbol{\xi}) = \frac{1}{2\pi}(\Gamma_4\xi_9 - A_2^4 + \Omega_2)$$

and so the critical point equations on $\mathbb{R}^9$ are $F_{d,i}(\boldsymbol{\xi}) = 0$, $i \in [9]$. That is,

$$\frac{1}{2\pi}(\Gamma_1\xi_1 - A_1^1 + \Omega_1) = 0 \qquad \frac{1}{2\pi}(\Gamma_1\xi_2 - A_\star^1 + \Omega_1) = 0$$

$$\frac{1}{2\pi}(\Gamma_1\xi_3 - A_2^1 + \Omega_2) = 0$$

$$\frac{1}{2\pi}(\Gamma_2\xi_4 - A_2^1 + \Omega_1) = 0 \qquad \frac{1}{2\pi}(\Gamma_2\xi_5 - A_2^2 + \Omega_2) = 0$$

$$\frac{1}{2\pi}(\Gamma_3\xi_6 - A_1^3 + \Omega_1) = 0 \qquad \frac{1}{2\pi}(\Gamma_3\xi_7 - A_2^3 + \Omega_2) = 0$$

$$\frac{1}{2\pi}(\Gamma_4\xi_8 - A_1^4 + \Omega_1) = 0 \qquad \frac{1}{2\pi}(\Gamma_4\xi_9 - A_2^4 + \Omega_2) = 0$$

If $\boldsymbol{\xi} \in \mathbb{R}^9$ determines a type II critical point, then $\xi_1 \to +1$ as $d \to \infty$. Numerics also indicate that for type II one of $\xi_5, \xi_7, \xi_9$ converges to $-1$ as $d \to \infty$. Permuting with an element of $S_3$, we may and shall hypothesize that $\xi_9 \to -1$ as $d \to \infty$. It helps to use some geometry concerning the rows $\boldsymbol{w}_d, \boldsymbol{w}_{d+1}$ of $W = \Xi(\boldsymbol{\xi})$. Let $\boldsymbol{u}_d = \sum_{i \in [d-1]} \boldsymbol{v}_i \in \mathbb{R}^d$ and $\mathbb{F} \subset \mathbb{R}^d$ be the 2-dimensional subspace spanned by $\boldsymbol{v}_d, \boldsymbol{u}_d$. Observe that $\boldsymbol{w}_d, \boldsymbol{w}_{d+1}, \boldsymbol{w}_k, \boldsymbol{v}_d \in \mathbb{F}$ and so are coplanar. By the analyticity properties of regular families, $\boldsymbol{w}_d$ and $\boldsymbol{w}_{d+1}$ cannot be parallel. Hence either $\beta_2 < \lambda_{32}$ or $\beta_2 > \lambda_{32}$: *curves of regular families do not cross*. In the first case $\Lambda_{23} = \lambda_{32} - \beta_2$; in the second $\Lambda_{23} = \beta_2 - \lambda_{32}$. Composing with a unique $\sigma \in S_3$, fixing the last row, we may always assume $\Lambda_{23} = \lambda_{32} - \beta_2$. Numerics indicate that $\lambda_{32} > \beta_2 > \pi/2 > \Lambda_{23} > 0$—but this is not assumed in what follows.

The idea now is to take formal FPS expansions for the components of a type II critical point, substitute in the critical point equations described above, equate like coefficients and thereby obtain FPS solutions. Guided by the numerics, we seek a power series in $d^{-\frac{1}{2}}$. Granted our hypothesis on the constant terms in the FPS, knowledge of vanishing coefficients (see Section A.5), and an easy computation comparing like terms giving $c_{3,2}$, we have

$$c_{1,0} = 1, \ c_{1,i} = 0, \ i \in [2], \ c_{2,i} = 0, \ i \le 3, \ c_{3,i} = 0, \ i < 2, \ c_{3,2} = 2,$$
$$c_{4,i} = 0, \ i < 2, \ c_{5,0} = 0, \ c_{6,i} = 0, i < 2, \ c_{7,0} = 0, \ c_{8,i} = 0, \ i < 2,$$
$$c_{9,0} = -1.$$

For notational clarity, we relabel the 9 unknown coefficients $c_{i,j}$ giving the next terms in the FPS so that we aim to find $c_3, e_4, f_3, g_2, h_1, p_2, q_1, a_2, b_1 \in \mathbb{R}$ such that

$$\xi_1 = 1 + c_3 d^{-\frac{3}{2}} + \cdots \quad \xi_2 = e_4 d^{-2} + \cdots \quad \xi_3 = 2d^{-1} + f_3 d^{-\frac{3}{2}} \cdots$$
$$\xi_4 = g_2 d^{-1} + \cdots \qquad \xi_5 = h_1 d^{-\frac{1}{2}} + \cdots$$
$$\xi_6 = p_2 d^{-1} + \cdots \qquad \xi_7 = q_1 d^{-\frac{1}{2}} + \cdots$$
$$\xi_8 = a_2 d^{-1} + \cdots \qquad \xi_9 = -1 + b_1 d^{-\frac{1}{2}} + \cdots$$

The condition we gave on angles holds if and only if $h_1 p_2 > g_2 q_1$ (both sides are negative). Set $R_2 = \sqrt{g_2^2 + h_1^2}$, $R_3 = \sqrt{p_2^2 + q_1^2}$. We derive expressions for the angles $\Lambda_{ab}$, $a, b \in \{2, 3, 4\}$ and find that

$$\Lambda_{34} = \Lambda_{34}^0 + O(d^{-\frac{1}{2}}), \ \ \Lambda_{23} = \Lambda_{23}^0 + O(d^{-\frac{1}{2}}), \ \ \Lambda_{24} = \Lambda_{24}^0 + O(d^{-\frac{1}{2}}),$$

where

$$\Lambda_{24}^0 = \sin^{-1}(g_2/R_2) \in (0, \pi/2)$$
$$\Lambda_{23}^0 = \sin^{-1}((h_1 p_2 - g_2 q_1)/(R_2 R_3)) \in (0, \pi/2) \text{ (since } h_1 p_2 > g_2 q_1)$$
$$\Lambda_{34}^0 = \sin^{-1}(p_2/R_3) \in (0, \pi/2)$$

Using standard trigonometric formulas, we deduce the relationship $\Lambda_{23} + \Lambda_{34} = \Lambda_{24}$ and, letting $d \to \infty$, $\Lambda_{23}^0 + \Lambda_{34}^0 = \Lambda_{24}^0$. We have similar expressions for $\beta_2$ and $\lambda_{a2}$, $a \in \{3, 4\}$:

$$\beta_2 = \beta_2^0 + O(d^{-\frac{1}{2}}) \quad \lambda_{32} = \lambda_{32}^0 + O(d^{-\frac{1}{2}}) \quad \lambda_{42} = \lambda_{42}^0 + O(d^{-\frac{1}{2}})$$

$$\beta_2^0 = \cos^{-1}(h_1/R_2) \in (\pi/2, \pi) \quad \lambda_{32}^0 = \cos^{-1}(q_1/R_3) \in (\pi/2, \pi) \quad \lambda_{42}^0 = \pi$$

It follows that $\Lambda_{24} + \beta_2 = \lambda_{42}$ and $\Lambda_{34} + \lambda_{32} = \lambda_{42}$, with the same identities holding between the constant terms by letting $d \to \infty$. In particular, all the constant terms for the FPS expansions of these angles can be expressed in terms of $R_2, \Lambda_{24}^0$ and $R_3, \Lambda_{34}^0$ (polar coordinates on $(g_2, h_1)$- and $(p_2, q_1)$-space).

Equating like coefficients of terms in the equations and noting in particular that the coefficients $\pi(e_4 + g_2 + p_2 + a_2)$ of $d^{-1}$ in $\Omega_1$, and $\pi(f_3 + h_1 + p_1 + b_1)$ of $d^{-\frac{1}{2}}$ in $\Omega_2$ are zero, we derive a

system of nine nonlinear equations that determine $c_3, e_4, f_3, g_2, h_1, p_2, q_1, a_2, b_1$:

$$c_3 - b_1 + R_2 + R_3 = 0$$
$$e_4 + g_2 + p_2 + a_2 = 0$$
$$f_3 + h_1 + q_1 + b_1 = 0$$
$$g_2\left(\frac{c_3 R_2^2 - 2g_2 h_1}{R_2^3} + \frac{-b_1 g_2 + h_1 a_2 - g_2 q_1 + h_1 p_2}{R_2^2}\right) = \Lambda_{23}^0 p_2 + \Lambda_{24}^0 a_2 + \frac{\pi}{2}e_4 - \frac{2h_1}{R_2}$$
$$h_1\left(\frac{c_3 R_2^2 - 2g_2 h_1}{R_2^3} + \frac{-b_1 g_2 + h_1 a_2 - g_2 q_1 + h_1 p_2}{R_2^2}\right) = \Lambda_{23}^0 q_1 + \Lambda_{24}^0 b_1 + \frac{\pi}{2}f_3 - \frac{2g_2}{R_2} + a_2$$
$$p_2\left(\frac{c_3 R_3^2 - 2p_2 q_1}{R_3^3} + \frac{-b_1 p_2 + q_1 a_2 + p_2 h_1 - q_1 g_2}{R_3^2}\right) = \Lambda_{23}^0 g_2 + \Lambda_{24}^0 a_2 + \frac{\pi}{2}e_4 - \frac{2q_1}{R_3}$$
$$q_1\left(\frac{c_3 R_3^2 - 2p_2 q_1}{R_3^3} + \frac{-b_1 p_2 + q_1 a_2 + p_2 h_1 - q_1 g_2}{R_3^2}\right) = \Lambda_{23}^0 h_1 + \Lambda_{24}^0 b_1 + \frac{\pi}{2}f_3 - \frac{2p_2}{R_3} + a_2$$
$$2 - \frac{\pi}{2}e_4 = (\pi - \Lambda_{24}^0)g_2 + (\pi - \Lambda_{34}^0)p_2 + \pi a_2$$
$$c_3 - \frac{\pi}{2}f_3 + g_2 + p_2 = (\pi - \Lambda_{24}^0)h_1 + (\pi - \Lambda_{34}^0)q_1 + \pi b_1$$

It is possible to reduce to a system of four equations in $R_2, \Lambda_{24}^0, R_3, \Lambda_{34}^0$ (or two in $\Lambda_{24}^0, \Lambda_{34}^0$), solve and then express the coefficients $c_3, e_4, \cdots, b_1$ in terms of $R_2, \Lambda_{\ell k}^0, R_3, \Lambda_{dk}^0$. In practice, we either use Newton-Raphson method on the original system, using an initialization based on numerics, or eliminate $c_3, e_4, f_3$ from the 9 equations and solve the resulting nonlinear system of 6 equations using Newton-Raphson. We used the reduction to a system of six equations, and found that

$$
\begin{array}{llll}
c_3 & = & -0.5748287640041448964\ldots & e_4 & = & -1.6165352425422284608\ldots \\
f_3 & = & 0.2969965493462016520\ldots \\
g_2 & = & 0.7877659431796313120\ldots & h_1 & = & -1.1161365378487412475\ldots \\
p_2 & = & 0.1562694812799615923\ldots & q_1 & = & -0.4248280138040598900\ldots \\
a_2 & = & 0.6724998180826355564\ldots & b_1 & = & 1.2439680023065994855\ldots
\end{array}
$$

Newton's method was initialized using numerical data for the critical points when $d = 10^4$ to get rough estimates for the coefficients; the original computation was done in long double precision. The values were compared with those obtained by directly solving FPS equations corresponding to different orders of the FPS terms (see Section A.5), and through a high precision computation of the critical point for $d = 10^{512}$—the values of the components of the gradient at the critical point were all less than $10^{-4000}$ and matched with those obtained by solutions of the equations above. Note that for $d = 10^{512}$, one expects to be able to read off the required coefficients from the computed critical points to 250 or more decimal places of accuracy (the series is in integer powers of $d^{-\frac{1}{2}}$).

**Remark 4** *It is straightforward to check that the Jacobian of the 9 variable system in $(c_3, \cdots, b_1)$ with respect to $(c_3, \cdots, b_1)$ is non-singular (for this, we need expressions for the angles $\Lambda_{ab}^0$, $a, b \in \{2, 3, 4\}$, $a \neq b$, in terms of $(c_3, \cdots, b_1)$). This is significant for the next and final step.*

Having determined $c_3, e_4, \cdots, b_1$, we set $t^{-\frac{1}{2}} = s$ and define $\widetilde{\xi}_i(s)$, $i \in [9]$ by

$$\xi_1(s) = -1 + s^3\widetilde{\xi}_1(s), \ \xi_2(s) = s^4\widetilde{\xi}_2(s), \ \xi_1(s) = 2s^{-1} + s^3$$
$$\xi_4(s) = s^2\widetilde{\xi}_4(s), \ \xi_5(s) = s\widetilde{\xi}_5(s)$$
$$\xi_6(s) = s^2\widetilde{\xi}_6(s), \ \xi_7(s) = s\widetilde{\xi}_5(s)$$
$$\xi_8(s) = s^2\widetilde{\xi}_8(s), \ \xi_9(s) = -1 + s\widetilde{\xi}_9(s)$$

where the values of $\widetilde{\xi}_1(0), \ldots, \widetilde{\xi}_9(0)$ are given by $(c_3, \cdots, b_1)$ respectively. Substitute in the critical point equations—with $d$ everywhere replaced by $s^{-2}$—and cancel the powers of $s$ that occur in each equation. The Jacobian of this system with respect to $(\widetilde{\xi}_1, \ldots, \widetilde{\xi}_9)$ at $s = 0$ is non-singular (this uses Remark 4). Finally, use the real analytic version of the implicit function theorem to obtain the FPS.

**Remark 5** *Although the argument only gives the convergence of the FPS for sufficiently large $d$ (that is, sufficiently small $d^{-\frac{1}{2}}$), it appears that the series converges for small $d$, possibly all $d$ for which the problem is defined. This is similar to what happens when $k = d$ [30].*

## A.2 Type II, $k = d + 1$

If $k = d + 1$, we can reduce the analysis to solving a single equation $p(\vartheta) = 0$, where $p$ is a polynomial in $\vartheta, \sin(\vartheta)$ and $\cos(\vartheta)$. We solve $p = 0$ directly using Newton's method, initializing with a value of $\vartheta$ suggested by numerics. For completeness, we give $p$ explicitly as well as the coefficients of interest in the associated FPS.

We have $\dim(M(d+1,d)^{\Delta S_{d-1}}) = 7$. The critical point equations are read off easily from those we gave for $k = d + 2$: drop the last two equations and all terms that involve the variables $\xi_8, \xi_9$ or an angle indexed with a '4'. Just as in the case when when $k = d + 2$, we reduce to finding the initial coefficients $c_3, e_4, f_3, g_2, h_1, p_2, q_1$ in the FPS expansion

$$\xi_1 = 1 + c_3 d^{-\frac{3}{2}} + \cdots, \quad \xi_2 = e_4 d^{-2} + \cdots, \quad \xi_3 = 2d^{-1} + f_3 d^{-\frac{3}{2}} + \cdots,$$

$$\xi_4 = g_2 d^{-1} + \cdots, \quad \xi_5 = h_1 d^{-\frac{1}{2}} + \cdots$$

$$\xi_6 = p_2 d^{-1} + \cdots, \quad \xi_7 = -1 + q_1 d^{-\frac{1}{2}} + \cdots$$

As in the case $k = d + 2$, we obtain a system of seven nonlinear equations.

$$c_3 - b_1 + R_2 = 0$$

$$e_4 + g_2 + a_2 = 0$$

$$f_3 + h_1 + b_1 = 0$$

$$g_2 \left( \frac{c_3 R_2^2 - 2g_2 h_1}{R_2^{\frac{3}{2}}} + \frac{-b_1 g_2 + h_1 a_2}{R_2^2} \right) = \vartheta a_2 + \frac{\pi}{2} e_4 - \frac{2h_1}{R_2}$$

$$h_1 \left( \frac{c_3 R_2^2 - 2g_2 h_1}{R_2^{\frac{3}{2}}} + \frac{-b_1 g_2 + h_1 a_2}{R_2^2} \right) = \frac{\pi}{2} f_3 - \frac{2g_2}{R_2} + \vartheta b_1 + a_2$$

$$\frac{\pi}{2} e_4 + 2 + \vartheta g_2 = 0$$

$$c_3 + \frac{\pi}{2} f_3 + g_2 + \vartheta h_1 = 0,$$

where $R_2 = \sqrt{g_2^2 + h_1^2}$ and $\vartheta = \Lambda_{23}^0 = \sin^{-1}\left(\frac{g_2}{R_2}\right) \in (0, \frac{\pi}{2})$.

We may reduce to a single equation $p(\vartheta) = 0$, where $p = AQ - BP$ and

$$A(\vartheta) = \frac{2}{2-\pi} \left[ \frac{\sin(2\vartheta)}{2}(1 - \sin(\vartheta))(\vartheta - \frac{\pi}{2}) + \sin(\vartheta)(1 - \sin(\vartheta))^2 \right]$$

$$+ \sin(\theta) \left[ 2\theta - \frac{2\theta^2}{\pi} - 1 - \frac{\sin(2\theta)}{2}(\frac{2\theta}{\pi} - 1) \right]$$

$$B(\vartheta) = \frac{2}{2-\pi} \left[ -\cos(\vartheta)(\frac{\pi}{2} - \vartheta)^2 + (\frac{\pi}{2} - \vartheta)(1 - \sin(\vartheta))(2 - \sin^2 \vartheta) - \cos(\vartheta)(1 - \sin(\vartheta))^2 \right]$$

$$+ \left[ \cos(\vartheta)(1 - \frac{\pi}{2}) - \sin^3 \vartheta(\frac{2\vartheta}{\pi} - 1) \right]$$

$$P(\vartheta) = 2 - \frac{4\vartheta}{\pi} - \frac{2\sin(2\vartheta)}{\pi} - 2\cos^3 \vartheta$$

$$Q(\vartheta) = 2\sin^3 \vartheta - \frac{4}{\pi}\sin^2 \vartheta$$

Using Newton-Raphson, the required solution to $p(\vartheta) = 0$ is given by

$\vartheta = 0.58416413506022510436594641534260755532740719514252671834097577387592202\ldots$

with $|p'(\vartheta)| > 0.4$ and $|p(\vartheta)| < 10^{-2400}$

We have $g_2 = R_2 \sin(\vartheta)$, $h_1 = -R_2 \cos(\vartheta)$ and $R_2 = -P(\vartheta)/A(\vartheta)$. It follows easily that the remaining coefficients $c_3, e_2, \cdots, b_1$ are uniquely determined by $\vartheta$. We find that

| | | | |
|---|---|---|---|
| $c_3 =$ | $-0.57228787893585490607\ldots$ | $e_4 =$ | $-1.61458989052095508224\ldots$ |
| $f_3 =$ | $0.29629854644604431015\ldots$ | | |
| $g_2 =$ | $0.91787878976036618322\ldots$ | $h_1 =$ | $-1.38833511087258399162\ldots$ |
| $a_2 =$ | $0.69671110076058889902\ldots$ | $b_1 =$ | $1.09203656442653968159\ldots$ |

The existence of the FPS now follows the method for $k = d + 2$.

## A.3  Type I, $k = d+1, d+2$

We conclude with a brief discussion of the type I family when $k = d+1, d+2$. This behaves rather differently from type II. For example, no row of a critical point converges to $v_d$ as $d \to \infty$. Instead, one row converges to $v_d/2$, another to $-v_d/2$. Moreover, the fractional power series are in $d^{-\frac{1}{4}}$ rather than $d^{-\frac{1}{2}}$. As a result, the convergence of critical points as $d \to \infty$ is slow. If $k = d+1$, explicit expressions for the critical coefficients are given in Example 1 of Section 3 (see Section A.5 for a detailed derivation). The analysis then proceeds as in the type II case.

If $k = d+2$, we again have the constant coefficients $1/2, -1/2$ for row-types $2, 3$ respectively (if necessary after a permutation by an element of $S_3$). Here is probably easiest (certainly faster), to obtain the system of nine equations, as we did for type II, and then solve using Newton-Raphson. In brief, after some work the critical coefficients for the FPS in $d^{-\frac{1}{4}}$ are given by

$$
\begin{aligned}
c_6 &= 2.6472714633048307498\ldots & e_7 &= -1.0684533932698202809\ldots \\
f_5 &= -0.8644915139550179823\ldots \\
g_3 &= 0.5342266966349101404\ldots & h_1 &= 0.4322457569775089911\ldots \\
p_3 &= 0.5342266966349101404\ldots & q_1 &= 0.4322457569775088806\ldots \\
a_4 &= 1.2534701553854549462\ldots & b_1 &= -0.8753051450722888701\ldots
\end{aligned}
$$

Note that $c_4 = e_4 = 2$, $f_4 = 1$, $g_3 = p_3$ and $h_1 = q_1$, just as for $k = d+1$. The existence of the FPS follows as above.

## A.4  Hessian spectrum for type I

In Table 2 we give the Hessian spectrum associated to the standard and trivial representations for type I points when $k = d+1, d+2$ for $d = 10, 100$.

| Isotypic comp. | $k = d+1$ | $k = d+2$ | $k = d+1$ | $k = d+2$ |
|---|---|---|---|---|
| $d$ | 10 | 10 | 100 | 100 |
| $\mathfrak{t}$ | $0.01859, 0.03574$ $0.08472, 0.25925$ $1.5634, 2.90219$ $3.3678$ | $0.00613, 0.01715$ $0.04434, 0.05308$ $0.2309, 0.3080$ $1.6453, 3.2022$ $3.7551$ | $0.006647, 0.05914$ $0.20056, 0.27534$ $15.746, 25.4661$ $26.055$ | $0.006467, 0.01348$ $0.06335, 0.09132$ $0.2697, 0.4127$ $15.846, 25.74$ $26.395$ |
| $\mathfrak{s}_{d-1}$ | $-0.00230, 0.04343$ $0.1135, 0.2324$ $0.3915, 2.9398$ | $-0.03903, 0.00423$ $0.04824, 0.1206$ $0.2630, 0.4559$ $3.2410$ | $0.03178, 0.0680$ $0.09210, 0.24363$ $0.48426, 25.478$ | $-0.03432, 0.03707$ $0.07250, 0.09303$ $0.3103, 0.5329$ $25.751$ |

Table 2: Type I critical points with isotropy $\Delta S_{d-1}$ & the Hessian eigenvalues associated to the trivial and the standard representation to four decimal places when $k = d+1, d+2$ and $d = 10$ and $100$. The spectrum associated to the $\mathfrak{r}$- and $\mathfrak{y}$-representations is strictly positive and not shown.

Referring to the table, the addition of one extra neuron results in the type I critical point becoming a saddle when $d = 10$ (it defines a spurious minimum if $k = d = 10$) but is a spurious minimum for $d = 100$. If we add a second neuron, the type I critical point becomes a saddle for $d = 100$; most likely a saddle for all $d \geq 4$ (certainly sufficiently large $d$ by our results).

## A.5  Type of regular families and derivation of case (3) in Example 2

Evaluating formal Puiseux series at critical points gives rise to algebraic relations between the Puiseux series coefficients, allowing one to argue about the structure of regular families (a-priori, independently of their existence). We show how these relations can be used to deduce that for $k = d+1$, any regular family of $\Delta(S_{d-1} \times S_1)$-critical points with $b = 4$ must be either type I, type II or have all its initial terms vanish. Other isotropy types and pairs of $d$ and $k$ are addressed similarly. Although only the diagonal of the main $(d-1) \times (d-1)$-block is required for determining the type of a family (see Definition 2), we evaluate all the entries which belong to the $(d-1) \times d$-upper block to low-order terms. This makes the derivation of the eigenvalue expressions given in (Section B.1)

more transparent, and further serves as a preparation for the detailed derivation below for the type I $\Delta(S_{d-1} \times S_1)$-critical points given in Example 2.

As in the previous sections, denote the FPS coefficients corresponding to $\xi_1, \xi_2, \ldots, \xi_7$ by $c_i, e_i, f_i, g_i, h_i, p_i, q_i$, and let $F_{d,i}$ denote the $i$'th component of the vector field $F$. Note that regularity assumptions imply that all coefficients with (strictly) negative index must vanish (e.g., necessarily, $c_{-1} = 0$). We show that for any regular family of critical points in $M(d+1, d)^{\Delta(S_{d-1} \times S_1)}$ with $b = 4$, it follows that $c_0 = \pm 1$ and $c_1 = e_0 = e_1 = e_2 = e_3 = f_0 = f_1 = f_2 = f_3 = 0$. The following notation comes handy when handling expressions involving Puiseux series coefficients. Given a Puiseux series $E = \sum_{j=j_0}^{\infty} \eta_j d^{-\frac{j}{4}}$ where $j_0 \in \mathbb{Z}$, let $[d^{-\frac{j}{4}}]E := \eta_j$.

Observe that

$$[d]F_{d,1} = \frac{e_0}{2} \quad \text{and} \quad [d]F_{d,3} = \frac{f_0}{2}.$$

Since the gradient entries vanish at critical points, necessarily, $e_0 = f_0 = 0$. Similarly,

$$[d^{\frac{3}{4}}]F_{d,1} = \frac{e_1}{2} \quad \text{and} \quad [d^{\frac{3}{4}}]F_{d,3} = \frac{f_1}{2},$$

imply $e_1 = f_1 = 0$. Assuming momentarily $c_0 \neq 0$, we have

$$[d]F_{d,1} = \frac{c_0\sqrt{-\frac{e_2^4}{c_0^4 + 2c_0^2 e_2^2 + e_2^4} + 1}}{2\pi} - \frac{c_0}{2\pi\sqrt{c_0^2 + e_2^2}},$$

$$[d^{\frac{1}{2}}]F_{d,2} = \frac{e_2\sqrt{-\frac{e_2^4}{c_0^4 + 2c_0^2 e_2^2 + e_2^4} + 1}}{2\pi} - \frac{e_2 \operatorname{acos}\left(\frac{e_2^2}{c_0^2 + e_2^2}\right)}{2\pi} + \frac{e_2}{2} - \frac{e_2}{2\pi\sqrt{c_0^2 + e_2^2}}.$$

Dividing the first expression by $c_0$ and using the resulting expression to simplify the second one gives

$$\frac{e_2}{2} - \frac{e_2 \operatorname{acos}\left(\frac{e_2^2}{c_0^2 + e_2^2}\right)}{2\pi} = 0.$$

If, by way of contradiction, $e_2$ is assumed to be non-zero then the preceding equation gives

$$\operatorname{acos}\left(\frac{e_2^2}{c_0^2 + e_2^2}\right) = \pi,$$

a contradiction, as the argument of $\arccos$, $\frac{e_2^2}{c_0^2 + e_2^2}$, is nonnegative. Hence, $e_2 = 0$ and

$$[d]F_{d,1} = c_0\left(\frac{1}{2\pi} - \frac{1}{2\pi|c_0|}\right),$$

whence $c_0 = \pm 1$. Evaluating $[d^{\frac{3}{4}}]F_{d,1}$, $[d^{\frac{1}{2}}]F_{d,3}$, $[d^{\frac{1}{4}}]F_{d,2}$ and $[d^{\frac{1}{4}}]F_{d,3}$ then yields $c_1 = e_3 = f_2 = f_3 = 0$, as required.

The case $c_0 = 0$ is addressed similarly. We shall only point out a possible course of derivation rather than provide the full expressions. Recall that $e_0 = e_1 = f_0 = f_1 = 0$ holds regardless of the value assigned to $c_0$. Evaluating $[d^{\frac{1}{2}}]F_{d,2}$ and $[d^{\frac{1}{2}}]F_{d,3}$ gives $e_2 = f_2 = 0$. Evaluating $[d^{\frac{1}{4}}]F_{d,2}$ gives $e_3 = 0$. By $[d^{\frac{1}{2}}]F_{d,4}$ and $[d^{\frac{1}{2}}]F_{d,6}$, $g_0 = p_0 = 0$. Lastly, evaluating $[d]F_{d,5}$ and $[d]F_{d,7}$ gives $h_0 = q_0 = 0$, concluding the derivation.

The procedure just presented is based on the direct approach described in [30, Section 8] by which one extracts coefficients by directly solving the FPS equations, exactly or numerically, to an increasing order. One proceeds until sufficient information has been obtained so as to establish the existence of an FPS and estimate the Hessian spectrum to a desired order. Below we give a detailed derivation of the type I $\Delta(S_{d-1} \times S_1)$-minima in Example 2 to demonstrate how the approach may be used in practice.

**A detailed derivation of case (3) in Example 2.** (Notation and assumption as above.) For any family of $\Delta(S_{d-1} \times S_1)$-critical points, $c_0 = \pm 1$ and $c_1 = e_0 = e_1 = e_2 = e_3 = f_0 = f_1 = f_2 = f_3 = 0$. The family of critical points given in Example 2 is type I, and so $c_0 = -1$. The derivation proceeds as follows.

1. Observe that

$$[d^0]F_{d,2} = \frac{e_4}{4} + \frac{g_0}{4} + \frac{p_0}{4} - \frac{1}{2},$$
$$[d^0]F_{d,3} = \frac{f_4}{4} + \frac{h_0}{4} + \frac{q_0}{4} - \frac{1}{4}.$$

We use these relations to substitute $e_4$ and $f_4$ for lower-order terms. That is,

$$e_4 = -g_0 - p_0 + 2,$$
$$f_4 = -h_0 - q_0 + 1.$$

2. By $[d^{\frac{1}{2}}]F_{d,1} = \frac{c_2}{2\pi} - \frac{|g_0|}{2\pi} - \frac{|p_0|}{2\pi}$, $c_2 = |g_0| + |p_0|$.

3. We now show that $g_0 = p_0 = 0$. If $g_0 = 0$ (resp. $p_0 = 0$) then by $[d^0]F_{d,6} = p_0 \left( \frac{1}{4} - \frac{1}{2\pi} \right)$ (resp. $[d^0]F_{d,4} = q_0 \left( \frac{1}{4} - \frac{1}{2\pi} \right)$), $p_0 = 0$ (resp. $g_0 = 0$). Assume then, by way of contradiction, that both $g_0$ and $p_0$ are non-zero, i.e., $g_0 \neq 0$ and $p_0 \neq 0$. Then $g_0$ and $p_0$ must satisfy the following four equations (effectively two, by symmetry) obtained by evaluating $[d^0]F_{d,4}, [d^0]F_{d,5}, [d^0]F_{d,6}$ and $[d^0]F_{d,7}$:

$$-\frac{g_0}{2\pi} + \frac{g_0}{4} - \frac{g_0|p_0|}{2\pi|g_0|} - \frac{p_0 \operatorname{acos}\left( \frac{g_0 p_0}{|g_0||p_0|} \right)}{2\pi} + \frac{p_0}{4} = 0, \tag{7}$$

$$-\frac{h_0}{2\pi} + \frac{h_0}{4} - \frac{h_0|p_0|}{2\pi|g_0|} - \frac{q_0 \operatorname{acos}\left( \frac{g_0 p_0}{|g_0||p_0|} \right)}{2\pi} + \frac{q_0}{4} = 0, \tag{8}$$

$$-\frac{g_0 \operatorname{acos}\left( \frac{g_0 p_0}{|g_0||p_0|} \right)}{2\pi} + \frac{g_0}{4} - \frac{p_0|g_0|}{2\pi|p_0|} - \frac{p_0}{2\pi} + \frac{p_0}{4} = 0, \tag{9}$$

$$-\frac{h_0 \operatorname{acos}\left( \frac{g_0 p_0}{|g_0||p_0|} \right)}{2\pi} + \frac{h_0}{4} - \frac{q_0|g_0|}{2\pi|p_0|} - \frac{q_0}{2\pi} + \frac{q_0}{4} = 0. \tag{10}$$

If $g_0, p_0 > 0$ or $g_0, p_0 < 0$, then by

$$[d^0]F_{d,4} = -\frac{g_0}{2\pi} + \frac{g_0}{4} - \frac{p_0}{2\pi} + \frac{p_0}{4},$$

it follows that $g_0 = -p_0$, a contradiction. If $p_0 < 0 < g_0$ then by $[d^0]F_{d,4}$ again, $p_0 = g_0$, still a contradiction. The case where $g_0 < 0 < p_0$ is treated similarly (and in fact follows by symmetry). Thus, necessarily, $g_0 = p_0 = 0$, and so $c_2 = 0$ and $e_4 = 2$.

4. Next, we have $[d^{\frac{1}{4}}]F_{d,1} = \frac{c_3}{2\pi} - \frac{|g_1|}{2\pi} - \frac{|p_1|}{2\pi}$ and $[d^{-\frac{1}{4}}]F_{d,2} = \frac{e_5}{4} + \frac{g_1}{4} + \frac{p_1}{4}$. Solving for $c_3$ and $e_5$ shows, by the same argument used in (3), that any of the four cases concerning the (strict) signs of $g_1$ and $p_1$ yields a contradiction. Thus, $g_1 = p_1 = 0$, and so $c_3 = e_5 = 0$.

5. We have $[d^0]F_{d,1} = \frac{c_4}{2\pi} - \frac{\sqrt{g_2^2 + h_0^2}}{2\pi} - \frac{\sqrt{p_2^2 + q_0^2}}{2\pi} - \frac{1}{2\pi}$. In addition, by $[d^{-\frac{1}{4}}]F_{d,3}$ we have $f_5 = -h_1 - q_1$, and by $[d^{-\frac{1}{2}}]F_{d,2}, e_6 = -g_2 - p_2$.

6. Using $c_4 = \sqrt{g_2^2 + h_0^2} + \sqrt{p_2^2 + q_0^2} + 1$, we obtain two (effectively one by symmetry) equations corresponding respectively to $[d^0]F_{d,5}$ and $[d^0]F_{d,7}$:

$$\text{(I)} \quad 0 = -\frac{h_0}{2\pi} + \frac{h_0}{4} - \frac{h_0|g_2|}{2\pi\,(g_2^2 + h_0^2)} + \frac{h_0|g_2 p_2 + h_0 q_0|}{2\pi\,(g_2^2 + h_0^2)} - \frac{h_0\sqrt{p_2^2 + q_0^2}}{2\pi\sqrt{g_2^2 + h_0^2}} + \frac{h_0}{2\pi\sqrt{g_2^2 + h_0^2}}$$

$$- \frac{q_0 \operatorname{acos}\left(\frac{g_2 p_2}{\sqrt{g_2^2+h_0^2}\sqrt{p_2^2+q_0^2}} + \frac{h_0 q_0}{\sqrt{g_2^2+h_0^2}\sqrt{p_2^2+q_0^2}}\right)}{2\pi} + \frac{q_0}{4} + \frac{\operatorname{acos}\left(\frac{h_0}{\sqrt{g_2^2+h_0^2}}\right)}{2\pi} - \frac{1}{4},$$

$$\text{(II)} \quad 0 = -\frac{h_0 \operatorname{acos}\left(\frac{g_2 p_2}{\sqrt{g_2^2+h_0^2}\sqrt{p_2^2+q_0^2}} + \frac{h_0 q_0}{\sqrt{g_2^2+h_0^2}\sqrt{p_2^2+q_0^2}}\right)}{2\pi} + \frac{h_0}{4} - \frac{q_0\sqrt{g_2^2 + h_0^2}}{2\pi\sqrt{p_2^2 + q_0^2}} - \frac{q_0}{2\pi} + \frac{q_0}{4}$$

$$- \frac{q_0|p_2|}{2\pi\,(p_2^2 + q_0^2)} + \frac{q_0|g_2 p_2 + h_0 q_0|}{2\pi\,(p_2^2 + q_0^2)} + \frac{q_0}{2\pi\sqrt{p_2^2 + q_0^2}} + \frac{\operatorname{acos}\left(\frac{q_0}{\sqrt{p_2^2+q_0^2}}\right)}{2\pi} - \frac{1}{4}.$$

In addition, by $[d^{-\frac{1}{2}}]F_{d,4}$,

$$\text{(III)} \quad 0 = -\frac{g_2}{2\pi} + \frac{g_2}{4} - \frac{g_2|g_2|}{2\pi\,(g_2^2 + h_0^2)} + \frac{g_2|g_2 p_2 + h_0 q_0|}{2\pi\,(g_2^2 + h_0^2)} - \frac{g_2\sqrt{p_2^2 + q_0^2}}{2\pi\sqrt{g_2^2 + h_0^2}} + \frac{g_2}{2\pi\sqrt{g_2^2 + h_0^2}}$$

$$- \frac{p_2 \operatorname{acos}\left(\frac{g_2 p_2}{\sqrt{g_2^2+h_0^2}\sqrt{p_2^2+q_0^2}} + \frac{h_0 q_0}{\sqrt{g_2^2+h_0^2}\sqrt{p_2^2+q_0^2}}\right)}{2\pi} + \frac{p_2}{4}.$$

The system of the FPS equations is symmetric under $(g_2, h_0) \leftrightarrow (p_2, q_0)$, and so we get the following symmetrized version of (III),

$$\text{(IV)} \quad 0 = -\frac{g_2 \operatorname{acos}\left(\frac{g_2 p_2}{\sqrt{g_2^2+h_0^2}\sqrt{p_2^2+q_0^2}} + \frac{h_0 q_0}{\sqrt{g_2^2+h_0^2}\sqrt{p_2^2+q_0^2}}\right)}{2\pi} + \frac{g_2}{4} - \frac{p_2\sqrt{g_2^2 + h_0^2}}{2\pi\sqrt{p_2^2 + q_0^2}} - \frac{p_2}{2\pi}$$

$$+ \frac{p_2}{4} - \frac{p_2|p_2|}{2\pi\,(p_2^2 + q_0^2)} + \frac{p_2|g_2 p_2 + h_0 q_0|}{2\pi\,(p_2^2 + q_0^2)} + \frac{p_2}{2\pi\sqrt{p_2^2 + q_0^2}}.$$

7. Our next goal is prove $p_2 = g_2 = 0$. This step is somewhat more involved. Recall that for the family described in Example 2 $h_0 = 1/2 = -q_0$, and so the following expressions are well-defined,

$$x = -\frac{1}{2\pi} + \frac{1}{4} - \frac{|g_2|}{2\pi\,(g_2^2 + h_0^2)} + \frac{|g_2 p_2 + h_0 q_0|}{2\pi\,(g_2^2 + h_0^2)} - \frac{\sqrt{p_2^2 + q_0^2}}{2\pi\sqrt{g_2^2 + h_0^2}} + \frac{1}{2\pi\sqrt{g_2^2 + h_0^2}},$$

$$y = -\frac{1}{2\pi} + \frac{1}{4} - \frac{|p_2|}{2\pi\,(p_2^2 + q_0^2)} + \frac{|g_2 p_2 + h_0 q_0|}{2\pi\,(p_2^2 + q_0^2)} - \frac{\sqrt{g_2^2 + h_0^2}}{2\pi\sqrt{p_2^2 + q_0^2}} + \frac{1}{2\pi\sqrt{p_2^2 + q_0^2}},$$

$$z = -\frac{\operatorname{acos}\left(\frac{g_2 p_2}{\sqrt{g_2^2+h_0^2}\sqrt{p_2^2+q_0^2}} + \frac{h_0 q_0}{\sqrt{g_2^2+h_0^2}\sqrt{p_2^2+q_0^2}}\right)}{2\pi} + \frac{1}{4}.$$

Equations (I-IV) now read:

$$\text{(I)} \quad 0 = h_0 x + q_0 z + \frac{\operatorname{acos}\left(\frac{h_0}{\sqrt{g_2^2+h_0^2}}\right)}{2\pi} - \frac{1}{4},$$

$$\text{(II)} \quad 0 = h_0 z + q_0 y + \frac{\operatorname{acos}\left(\frac{q_0}{\sqrt{p_2^2+q_0^2}}\right)}{2\pi} - \frac{1}{4},$$

$$\text{(III)} \quad 0 = g_2 x + p_2 z,$$

$$\text{(IV)} \quad 0 = g_2 z + p_2 y.$$

Combining (I) and (III) (resp. (II) and (IV)) by solving (I) for $x$ (resp. solving (II) for $y$) and substituting yields,

$$\text{(A)} \quad 0 = \frac{-g_2}{h_0} \left( q_0 z + \frac{\mathrm{acos}\left( \frac{h_0}{\sqrt{g_2^2 + h_0^2}} \right)}{2\pi} - \frac{1}{4} \right) + p_2 z,$$

$$\text{(B)} \quad 0 = g_2 z - \frac{p_2}{q_0} \left( h_0 z + \frac{\mathrm{acos}\left( \frac{q_0}{\sqrt{p_2^2 + q_0^2}} \right)}{2\pi} - \frac{1}{4} \right).$$

Summing (A) and (B) we have,

$$0 = \left( p_2 + g_2 - \frac{g_2 q_0}{h_0} - \frac{p_2 h_0}{q_0} \right) z - \frac{g_2}{h_0} \left( \frac{\mathrm{acos}\left( \frac{h_0}{\sqrt{g_2^2 + h_0^2}} \right)}{2\pi} - \frac{1}{4} \right) - \frac{p_2}{q_0} \left( \frac{\mathrm{acos}\left( \frac{q_0}{\sqrt{p_2^2 + q_0^2}} \right)}{2\pi} - \frac{1}{4} \right).$$

For the derivation so far to be valid only $h_0, q_0 \neq 0$ is required. Plugging-in $h_0 = 1/2 = -q_0$, the above becomes

$$0 = \frac{g_2 \, \mathrm{acos}\left( \frac{1}{2\sqrt{g_2^2 + \frac{1}{4}}} \right)}{\pi} - \frac{g_2}{2} - \frac{p_2 \, \mathrm{acos}\left( -\frac{1}{2\sqrt{p_2^2 + \frac{1}{4}}} \right)}{\pi} + \frac{p_2}{2}. \tag{11}$$

The function

$$f(x) = \frac{x \, \mathrm{acos}\left( \frac{1}{2\sqrt{x^2 + \frac{1}{4}}} \right)}{\pi} - \frac{x}{2}$$

is injective, and so $g_2 = p_2$ by Equation 11. Plugging in this into (A) yields $f(g_2) = 0$. Since $f(0) = 0$ we have, by the injectivity of $f$ again, that $g_2 = 0$, hence $h_2 = 0$ as well. Backward substitution then gives $c_4 = 2$, $f_4 = 1$, $e_6 = 0$. The FPS equations encountered in the reminder of the derivation are simpler.

8. We have $[d^{-\frac{1}{4}}]F_{d,5} = -\frac{h_1}{2\pi} + \frac{h_1}{4} - \frac{q_1}{4} + \frac{q_1}{2\pi}$, hence $h_1 = q_1$. Therefore, $[d^{-\frac{1}{4}}]F_{d,1} = \frac{c_5}{2\pi} - \frac{h_1}{2\pi} + \frac{q_1}{2\pi}$ implies $c_5 = 0$.

9. By $[d^{-\frac{1}{2}}]F_{d,3} = \frac{f_6}{4} + \frac{h_2}{4} + \frac{q_2}{4}$ and $[d^{-\frac{1}{2}}]F_{d,5} = -\frac{c_6}{2\pi} + \frac{f_6}{4} + \frac{h_2}{2}$, hence $f_6 = -h_2 - q_2$ and $c_6 = \frac{\pi h_2}{2} - \frac{\pi q_2}{2}$.

10. By $[d^{-\frac{3}{4}}]F_{d,2} = \frac{e_7}{4} + \frac{g_3}{4} + \frac{p_3}{4}$, $[d^{-\frac{3}{4}}]F_{d,3} = \frac{f_7}{4} + \frac{g_3}{2\pi} + \frac{h_3}{4} - \frac{p_3}{2\pi} + \frac{q_3}{4}$, $[d^{-\frac{3}{4}}]F_{d,4} = \frac{e_7}{4} + \frac{g_3}{2}$ and $[d^{-\frac{3}{4}}]F_{d,6} = \frac{e_7}{4} + \frac{p_3}{2}$, $e_7 = -2p_3$, $f_7 = -h_3 - j_3$ and $g_3 = p_3$.

11. By $[d^{-\frac{3}{4}}]F_{d,5}$, $c_7 = \frac{\pi h_3}{2} - 4p_3 - \frac{\pi q_3}{2}$, and by $[d^{-1}]F_{d,2}$, $e_8 = -g_4 - p_4 + \frac{4}{\pi}$.

12. We now have

$$[d^{-1}]F_{d,4} = \frac{g_4}{4} + \frac{2p_3|p_3|}{\pi} - \frac{p_4}{4} - \frac{1}{2} + \frac{3}{2\pi},$$

$$[d^{-1}]F_{d,6} = -\frac{g_4}{4} + \frac{2p_3|p_3|}{\pi} + \frac{p_4}{4} - \frac{1}{2} + \frac{1}{2\pi}.$$

Summing the two equations gives

$$\frac{4p_3|p_3|}{\pi} - 1 + \frac{2}{\pi} = 0.$$

The equation has a single root at $p_3 = g_3 = \frac{\sqrt{-2+\pi}}{2}$. Backward substitution then gives $e_7 = -\sqrt{-2 + \pi}$.

13. The expression $[d^{-\frac{1}{2}}]F_{d,1}$ depends on $g_3$ which has just been determined. Re-evaluating, we have $h_2 = q_2 + 1$, and so $c_6 = \frac{\pi}{2}$.

14. By $[d^{-1}]F_{d,4}$, $g_4 = p_4 - \frac{2}{\pi}$, by $[d^{-1}]F_{d,3}$, $f_8 = -h_4 - q_4 + \frac{4}{\pi^2}$, and by $[d^{-\frac{3}{4}}]F_{d,1}$,

$$h_3 = \frac{-4\pi p_4\sqrt{-2+\pi} - q_3(\pi^2 + 2\pi) + 4\sqrt{-2+\pi}}{\pi(2-\pi)}.$$

15. Now,

$$[d^{-1}]F_{d,5} = -\frac{c_8}{2\pi} + \frac{h_4}{4} - \frac{2p_4}{\pi} - \frac{4q_1\sqrt{-2+\pi}}{3\pi} + \frac{2q_1\sqrt{-2+\pi}}{3} - \frac{q_4}{4} - \frac{1}{2} - \frac{1}{2\pi} + \frac{\pi}{8} + \frac{7}{\pi^2},$$

$$[d^{-1}]F_{d,7} = \frac{c_8}{2\pi} - \frac{h_4}{4} + \frac{2p_4}{\pi} - \frac{4q_1\sqrt{-2+\pi}}{3\pi} + \frac{2q_1\sqrt{-2+\pi}}{3} + \frac{q_4}{4} - \frac{5}{\pi^2} - \frac{\pi}{8} + \frac{1}{2\pi}.$$

Summing the two equations above, we get

$$4q_1\sqrt{-2+\pi}\left(\frac{1}{3} - \frac{2}{3\pi}\right) - \frac{1}{2} + \frac{2}{\pi^2} = 0,$$

hence $q_1 = h_1 = \frac{6+3\pi}{8\pi\sqrt{-2+\pi}}$ (recall that $q_1 = h_1$). Consequently, $f_5 = -\frac{6+3\pi}{4\pi\sqrt{-2+\pi}}$.

16. The procedure may be further iterated by observing, e.g., that $[d^{-1}]F_{d,5}$ implies $c_8 = \frac{\pi h_4}{2} - 4p_4 - \frac{\pi q_4}{2} - \frac{\pi}{2} - 1 + \frac{\pi^2}{4} + \frac{12}{\pi}$ and so on, if additional coefficients are needed.

For type II critical points, the equations corresponding to Equations (I-IV) above are different, and we have not been able to solve them exactly. Rather, Newton-Raphson method was used to obtain numerical estimates. The same procedure was then applied iteratively, with the aid of numerical methods, giving estimates for higher-order terms. The estimates obtained through this process match with those obtained by the method described in Section A.1. Other choices of types, $k$, $d$ and isotropy were addressed similarly.

## A.6 Expected Initial Value

Bounding $\mathbb{E}_W[\mathcal{L}(W)]$ follows by a straightforward computation of the expected loss. We first derive explicit expressions for the terms used in computation.

$$\mathbb{E}_{(\boldsymbol{x},\boldsymbol{w})\sim\mathcal{N}(0,I_d)^{\otimes 2}}[\sigma^2(\langle \boldsymbol{w}, \boldsymbol{x}\rangle)] = \mathbb{E}_{\boldsymbol{w}\sim\mathcal{N}(0,I_d)}\frac{\|\boldsymbol{w}\|^2}{2} = \frac{d}{2},$$

$$\mathbb{E}_{\boldsymbol{x}\sim\mathcal{N}(0,I_d)}[\sigma(\langle \boldsymbol{v}_2, \boldsymbol{x}\rangle)\sigma(\langle \boldsymbol{v}_1, \boldsymbol{x}\rangle)] = \begin{cases} 1/2 & \boldsymbol{v}_1 = \boldsymbol{v}_2, \\ 1/(2\pi) & \boldsymbol{v}_1 \neq \boldsymbol{v}_2, \end{cases}$$

$$\begin{aligned}
\mathbb{E}_{\boldsymbol{w}\sim\mathcal{N}(0,I_d)}[\sigma(\langle \boldsymbol{w}, \boldsymbol{x}\rangle)] &= \frac{1}{2}\mathbb{E}_{\boldsymbol{w}\sim\mathcal{N}(0,I_d)}[\sigma(\langle \boldsymbol{w}, \boldsymbol{x}\rangle) \mid \langle \boldsymbol{w}, \boldsymbol{x}\rangle \geq 0] + \frac{1}{2}\mathbb{E}_{\boldsymbol{w}\sim\mathcal{N}(0,I_d)}[\sigma(\langle \boldsymbol{w}, \boldsymbol{x}\rangle) \mid \langle \boldsymbol{w}, \boldsymbol{x}\rangle < 0] \\
&= \frac{1}{2}\mathbb{E}_{\boldsymbol{w}\sim\mathcal{N}(0,I_d)}[\langle \boldsymbol{w}, \boldsymbol{x}\rangle \mid \langle \boldsymbol{w}, \boldsymbol{x}\rangle \geq 0] \\
&= \frac{1}{2}\mathbb{E}_{\boldsymbol{w}\sim\mathcal{N}(0,I_d)}[\boldsymbol{w} \mid \langle \boldsymbol{w}, \boldsymbol{x}\rangle \geq 0]^\top \boldsymbol{x} \\
&= \frac{\|\boldsymbol{x}\|}{\sqrt{2\pi}},
\end{aligned}$$

$$\begin{aligned}
\mathbb{E}_{\boldsymbol{w}\sim\mathcal{N}(0,I_d)}[\sigma(\langle \boldsymbol{w}, \boldsymbol{x}\rangle)] &= \frac{1}{2}\mathbb{E}_{\boldsymbol{w}\sim\mathcal{N}(0,I_d)}[\sigma(\langle \boldsymbol{w}, \boldsymbol{x}\rangle) \mid \boldsymbol{w}^\top \boldsymbol{x} \geq 0] + \frac{1}{2}\mathbb{E}_{\boldsymbol{w}\sim\mathcal{N}(0,I_d)}[\sigma(\langle \boldsymbol{w}, \boldsymbol{x}\rangle) \mid \langle \boldsymbol{w}, \boldsymbol{x}\rangle < 0] \\
&= \frac{1}{2}\mathbb{E}_{\boldsymbol{w}\sim\mathcal{N}(0,I_d)}[\langle \boldsymbol{w}, \boldsymbol{x}\rangle \mid \langle \boldsymbol{w}, \boldsymbol{x}\rangle \geq 0] \\
&= \frac{1}{2}\mathbb{E}_{\boldsymbol{w}\sim\mathcal{N}(0,I_d)}[\boldsymbol{w} \mid \langle \boldsymbol{w}, \boldsymbol{x}\rangle \geq 0]^\top \boldsymbol{x} \\
&= \frac{\|\boldsymbol{x}\|}{\sqrt{2\pi}},
\end{aligned}$$

$$\mathbb{E}_{\|\boldsymbol{x}\|=1}[\sigma(\langle \boldsymbol{v}, \boldsymbol{x}\rangle)] = \frac{\|\boldsymbol{v}\|}{2}\mathbb{E}_\theta[\langle \boldsymbol{v}/\|\boldsymbol{v}\|, \boldsymbol{x}\rangle \mid \langle \boldsymbol{v}/\|\boldsymbol{v}\|, \boldsymbol{x}\rangle \geq 0]$$

$$= \frac{\|\boldsymbol{v}\|}{2}\mathbb{E}_{\|\boldsymbol{x}\|=1}[x_1 \mid x_1 \geq 0]$$

$$= \frac{\|\boldsymbol{v}\|}{2}\frac{2}{d\mathrm{Beta}((n+1)/2, 1/2)}$$

$$= \frac{\|\boldsymbol{v}\|}{d\mathrm{Beta}((n+1)/2, 1/2)},$$

$$\mathbb{E}_{\substack{\boldsymbol{w}\sim\mathcal{N}(0,I_d)\\\boldsymbol{x}\sim\mathcal{N}(0,I_d)}}[\sigma(\langle \boldsymbol{w}, \boldsymbol{x}\rangle)\sigma(\langle \boldsymbol{v}, \boldsymbol{x}\rangle)]$$

$$= \mathbb{E}_{\boldsymbol{x}\sim\mathcal{N}(0,I_d)}\left[\frac{1}{\sqrt{2\pi}}\|\boldsymbol{x}\|\sigma(\langle \boldsymbol{v}, \boldsymbol{x}\rangle)\right]$$

$$= \mathbb{E}_r\mathbb{E}_{\|\boldsymbol{\theta}\|=1}\left[\frac{1}{\sqrt{2\pi}}r\|\boldsymbol{\theta}\|\sigma(\langle \boldsymbol{v}, r\boldsymbol{\theta}\rangle)\right]$$

$$= \mathbb{E}_r\frac{v}{\sqrt{2\pi}}r^2\mathbb{E}_{\|\boldsymbol{\theta}\|=1}[\sigma(\langle \boldsymbol{v}, \boldsymbol{\theta}\rangle)]$$

$$= \frac{1}{\sqrt{2\pi}}\frac{\|\boldsymbol{v}\|}{d\mathrm{Beta}((d+1)/2, 1/2)}\mathbb{E}_r r^2$$

$$= \frac{\|\boldsymbol{v}\|}{\sqrt{2\pi}\mathrm{Beta}((d+1)/2, 1/2)},$$

$$\mathbb{E}_{(\boldsymbol{x},\boldsymbol{v},\boldsymbol{w})\sim\mathcal{N}(0,I_d)^{\otimes 3}}[\sigma(\langle \boldsymbol{w}, \boldsymbol{x}\rangle)\sigma(\langle \boldsymbol{v}, \boldsymbol{x}\rangle)]$$

$$= \frac{\mathbb{E}_{\boldsymbol{v}}\|\boldsymbol{v}\|}{\sqrt{2\pi}\mathrm{Beta}((d+1)/2, 1/2)}$$

$$= \frac{1}{\sqrt{\pi}\mathrm{Beta}((d+1)/2, 1/2)}\frac{\Gamma((d+1)/2)}{\Gamma(d/2)}$$

$$= \frac{1}{\pi\mathrm{Beta}(d/2, 1)}$$

$$= \frac{d}{2\pi}.$$

Therefore,

$$\mathbb{E}_{\substack{\boldsymbol{x}\sim\mathcal{N}(0,I_d)\\W\sim\mathcal{N}(0_{d\times d}, I_{d^2})}}[(\mathbf{1}^\top\sigma\left(\frac{1}{\sqrt{d}}W\boldsymbol{x}\right) - \mathbf{1}^\top\sigma(V\boldsymbol{x}))^2]$$

$$= \mathbb{E}[d\frac{1}{d}\sigma^2(\langle \boldsymbol{w}, \boldsymbol{x}\rangle) + \frac{1}{d}d(d-1)\sigma(\langle \boldsymbol{w}_1, \boldsymbol{x}\rangle)\sigma(\langle \boldsymbol{w}_2, \boldsymbol{x}\rangle) - \frac{1}{\sqrt{d}}2d^2\sigma(\langle \boldsymbol{w}, \boldsymbol{x}\rangle)\sigma(\langle \boldsymbol{v}, \boldsymbol{x}\rangle) + d\sigma^2(\langle \boldsymbol{v}, \boldsymbol{x}\rangle)$$

$$+ d(d-1)\sigma(\langle \boldsymbol{v}_1, \boldsymbol{x}\rangle)\sigma(\langle \boldsymbol{v}_2, \boldsymbol{x}\rangle)]$$

$$= \frac{d}{2} + (d-1)\frac{d}{2\pi} - \frac{2d^{3/2}}{\sqrt{2\pi}\mathrm{Beta}((d+1)/2, 1/2)} + \frac{d}{2} + d(d-1)\frac{1}{2\pi}$$

$$= d + \frac{d(d-1)}{\pi} - \frac{\sqrt{2}d^{3/2}\Gamma(d/2+1)}{\sqrt{\pi}\Gamma((d+1)/2)\Gamma(1/2)}$$

$$\leq d + \frac{d(d-1)}{\pi} - \frac{\sqrt{2}d^{3/2}}{\pi}\left(\frac{d}{2}\right)^{1/2}$$

$$= \left(1 - \frac{1}{\pi}\right)d,$$

where the penultimate transition uses Gautschi's inequality. The same inequality also gives the following lower bound,

$$
\mathbb{E}[(\mathbf{1}^\top \sigma\left(\frac{1}{\sqrt{d}}W\boldsymbol{x}\right) - \mathbf{1}^\top \sigma(V\boldsymbol{x}))^2] \geq \left(1 - \frac{1}{\pi}\right)d + \frac{1}{\pi}(d^2 - \sqrt{2}d^{3/2}(d/2+1)^{1/2})
$$

$$
= \left(1 - \frac{1}{\pi}\right)d + \frac{d^2\left(1 - \sqrt{1 + \frac{2}{d}}\right)}{\pi}
$$

$$
\geq \left(1 - \frac{1}{\pi}\right)d + \frac{d^2(1 - (1 + 1/d))}{\pi}
$$

$$
= \left(1 - \frac{2}{\pi}\right)d,
$$

with the penultimate transition following by the first-order Taylor expansion of $\sqrt{1 + x}$.

### A.7 Adding more than two neurons: beyond non-degenerate critical points

When we add neurons to a shallow network, new critical points appear and old critical points become simplices with singular Hessian spectrum (at points where the Hessian is defined). This phenomenon is well-known and not restricted to ReLU networks. We shall refer to this process here as "fossilization", as the set of (connected) fossils together, with the discrete critical points, encodes information about the number of additional neurons involved, and how they were added. Thus the fossil record generated when $p > 1$ neurons are added simultaneously may be less informative than that generated when $p$ neurons are added one-by-one. Moreover, as we show, symmetry plays a significant role in the description of the fossilized sets; even if the target $V$ is asymmetric.

The critical points giving the global minimum fossilize when neurons are added. Recall that $\mathcal{L}$ is always $S_k$-invariant, where $S_k$ is the group of row permutations of $M(k, d)$. If $d = k$, we add the superscript $r$ (resp. $c$) to emphasize row (resp. column) permutations. In our setting, if $k = d$, $V = I_d$, the $d!$ points in the $S_d^r$-orbit of $V$ will be non-degenerate critical points of $\mathcal{L}$ giving the global minimum zero: these will be the only points in $M(d, d)$ that give the global minimum. If we add $p \geq 1$ neurons, the discrete $S_d^r$-orbit of $V$ is replaced by a $p$-dimensional connected $S_{d+p}$-invariant simplicial complex $Z \subset M(p + d, d)$ consisting of all points giving the global minimum. Necessarily, the Hessian, where defined on $Z$, will have zero eigenvalues; that does not preclude $Z$ from being an attractor under gradient descent. Suppose instead that $\mathfrak{c} \in M(k, d)$, $k \geq d$, is any (non-degenerate) critical point of $\mathcal{L}$. The addition of a $p$ neurons will replace $\mathfrak{c}$ by a connected $p$-dimensional simplex, invariant by the action of $S_d^r$. Often (not always) many new non-degenerate critical points will be created as biproducts of the fossilization process.

For completeness, we give precise statements and proof of these results, starting with the case when $\mathfrak{c} = V$ and $k = d$. We start with an extension of the result on the uniqueness of critical points defining global minima [30, Prop. 4.14] to the over-parameterized case. See [7] for related results and discussions.

Assume $k \geq d$, set $m = k - d$, and $\Gamma = S_k \times S_d$. Let $S_m = \{e_d\} \times S_m$ denote the subgroup of $S_k$ permuting the last $m$ rows of matrices in $M(k, d)$. Define $\Delta_m S_d = \{(hg, g) \mid gS_d, h \in S_m\}$. Note that $\Delta_m S_d \approx \Delta S_d \times S_m$.

Let $\mathfrak{K}^\star$ denote the set of all partitions of $[k]$ such that each $\mathcal{K} \in \mathfrak{K}^\star$ has exactly $d$ parts, $K_1, \cdots K_d$ and $j \in K_j$, for all $j \in [d]$. If $j \in [d]$, then

$$
K_j \cap [d] = \{j\}, \quad \text{and} \quad K_j \smallsetminus \{j\} \subset [k] \smallsetminus [d].
$$

Clearly, $1 \leq |K_j| \leq m + 1$ for all $j \in [d]$.

If $\mathcal{K} \in \mathfrak{K}^\star$, let $M_\mathcal{K} = [k_{ij}] \in M(k, d)$ be the matrix defined by

$$
k_{ij} = 0, \; i \notin K_j \tag{12}
$$
$$
= 1, \; i \in K_j \tag{13}
$$

For $j \in [d]$, define $\Delta_j(\mathcal{K}) \subset \mathbb{R}^k$ by

$$
\Delta_j(\mathcal{K}) = \{(t_1, \cdots, t_k) \in \prod_{i \in K_j}[0, k_{ij}] \mid \sum_{i \in K_j} t_i = 1\}
$$

and, viewing $M(k,d)$ as $(\mathbb{R}^k)^d$, define

$$\Delta(\mathcal{K}) = \prod_{j \in [d]} \Delta_j(\mathcal{K}) \subset M(k,d).$$

Clearly $\Delta(\mathcal{K})$ is a simplicial complex of dimension $m$ and if $m = 0$, $\Delta(\mathcal{K}) = \{V\}$. If $\boldsymbol{\delta} \in \Delta(\mathcal{K})$, then

$$\boldsymbol{\delta}^\Sigma = V^\Sigma = \mathcal{I}_{1,d},$$

where for $W \in M(k,d)$, $W^\Sigma$ is the $1 \times d$-row matrix defined by the column sums of $W$. Define

$$\boldsymbol{\Delta}(k,d) = \bigcup_{\mathcal{K} \in \mathfrak{K}^\star} \Delta(\mathcal{K}). \tag{14}$$

Suppose $W \in M(k,d)$ and $\boldsymbol{\delta} \in \Delta(\mathcal{K})$. Define $W_{\boldsymbol{\delta}} \in M(k,d)$ by

$$
\begin{aligned}
\boldsymbol{w}_i^{\boldsymbol{\delta}} &= \delta_{ii}\boldsymbol{w}_i, \ i \in [d] \\
&= \boldsymbol{w}_i + \sum_{j \in [d]} \delta_{ij}\boldsymbol{w}_j, \ i > d
\end{aligned}
$$

Observe that

$$W^\Sigma = W_{\boldsymbol{\delta}}^\Sigma.$$

The case of most interest will be when $W = V$ and so the last $m$ rows of $W$ will be zero.

**Lemma 2** *(Notation and assumptions as above.)*

1. *For all $\mathcal{K} \in \mathfrak{K}^\star$, $V \in \Delta(\mathcal{K})$.*

2. *If $\mathcal{K}, \mathcal{J} \in \mathfrak{K}^\star$, $\mathcal{K} \neq \mathcal{J}$, then $\Delta(\mathcal{K}) \cap \Delta(\mathcal{J})$ is a simplicial complex which is the union of the common vertices and faces of $\Delta(\mathcal{K})$ and $\Delta(\mathcal{J})$.*

3. *$\boldsymbol{\Delta}(k,d)$ is a connected simplicial complex of dimension $m$.*

4. *$\Delta_m S_d(\boldsymbol{\Delta}(k,d)) = \boldsymbol{\Delta}(k,d)$ and $\Delta_m S_d$ is the maximal subgroup of $\Gamma$ with this property.*

5. *If $g \in S_d^r \cup S_d^c$, $g \neq e$, then $g\boldsymbol{\Delta}(k,d) \cap \Gamma V = gV \neq V$.*

6. *$\mathcal{L}(W) = 0$ if $W \in \boldsymbol{\Delta}(k,d)$.*

Statements (1–3) are all immediate from the definitions and the proofs of (4,5) are straightforward and omitted. It remains to prove (6). Recall that

$$\mathcal{L}(W) = \frac{1}{2}\sum_{i,j \in [k]} f(\boldsymbol{w}_i, \boldsymbol{w}_j) - \sum_{i \in [k], j \in [d]} f(\boldsymbol{w}_i, \boldsymbol{v}_j) + \frac{1}{2}\sum_{i,j \in [d]} f(\boldsymbol{v}_i, \boldsymbol{v}_j),$$

where

1. If $\boldsymbol{v}, \boldsymbol{w} \in \mathbb{R}^d$ are non-zero and we set $\theta_{\boldsymbol{w},\boldsymbol{v}} = \cos^{-1}\left(\frac{\langle \boldsymbol{w}, \boldsymbol{v}\rangle}{\|\boldsymbol{w}\|\|\boldsymbol{v}\|}\right)$, then

$$f(\boldsymbol{w}, \boldsymbol{v}) = \frac{1}{2\pi}\|\boldsymbol{w}\|\|\boldsymbol{v}\|(\sin(\theta_{\boldsymbol{w},\boldsymbol{v}}) + (\pi - \theta_{\boldsymbol{w},\boldsymbol{v}})\cos(\theta_{\boldsymbol{w},\boldsymbol{v}}))$$

2. $f(\boldsymbol{w}, \boldsymbol{v}) = 0$ iff either $\boldsymbol{v}$ or $\boldsymbol{w}$ is zero or $\theta_{\boldsymbol{w},\boldsymbol{v}} = \pi$.

Clearly, $f$ is positively homogeneous:

$$f(a\boldsymbol{w}, b\boldsymbol{v}) = abf(\boldsymbol{w}, \boldsymbol{v}), \ a, b \geq 0, \ \boldsymbol{w}, \boldsymbol{v} \in \mathbb{R}^d.$$

If $W \in \boldsymbol{\Delta}(k,d)$, then there exist $\mathcal{K} \in \mathfrak{K}^\star$ and $\boldsymbol{\delta} \in \Delta(\mathcal{K})$ such that $W = V_{\boldsymbol{\delta}}$—$\boldsymbol{v}_j$ is zero for $j > d$. The result follows from the positive homogeneity of $f$ and the formula for $\mathcal{L}$ in terms of $f$. $\square$

**Remark 6** *If $k = d$, $V$ is the natural choice for a critical point on the group orbit of critical points giving the global minima. When $k > d$, the natural choice—at least from a symmetry perspective—is the set $\boldsymbol{\Delta}(k,d)$ which is invariant by $\Delta_m S_d$ (the isotropy group of $V$). It follows by (5) of the lemma that for all $g \in S_d \times S_d$, $g\boldsymbol{\Delta}(k,d)$ contains exactly one point in $\Gamma V$.*

Define $\boldsymbol{\Delta}^\star = \Gamma\boldsymbol{\Delta}(k,d)$ and note that $\Gamma[V] \subset \boldsymbol{\Delta}^\star$.

**Theorem 5** *(Notation and assumptions as above.)*

1. $\boldsymbol{\Delta}^\star$ *is a $\Gamma$-invariant $m$-dimensional simplicial complex of $M(k,d)$.*

2. $\boldsymbol{\Delta}^\star$ *is connected.*

3. $\mathcal{L}(W) = 0$ *iff* $W \in \boldsymbol{\Delta}^\star$.

By Lemma 2, $\boldsymbol{\Delta}(k,d)$ is a connected $m$-dimensional simplicial complex and it follows easily that $\boldsymbol{\Delta}^\star$ is a $\Gamma$-invariant $m$-dimensional simplicial complex, proving (1). (2) If $g \in \Gamma$, $g\boldsymbol{\Delta}(k,d) \cap \boldsymbol{\Delta}(k,d)$ may be empty. However, given any $g \in \Gamma$, it is easy to choose a sequence $g_1, \cdots g_{n+1} \in S_k$ such that such that $g_1 = e$, and $g_j\boldsymbol{\Delta}(k,d) \cap g_{j+1}\boldsymbol{\Delta}(k,d) \neq \emptyset$, $j \in [n]$, and $g_{n+1}\boldsymbol{\Delta}(k,d) \cap g\boldsymbol{\Delta}(k,d) \neq \emptyset$. Hence $\Gamma\boldsymbol{\Delta}(k,d) = \boldsymbol{\Delta}^\star$ is connected (see Example 3 below for more detail). It is straightforward to check that for all $g \in \Gamma$, and $\mathcal{K}, \mathcal{J} \in \mathfrak{K}^\star$, $g\Delta(\mathcal{K}) \cap \Delta(\mathcal{J})$ is a simplicial complex (possibly empty) and from this it follows that $\boldsymbol{\Delta}^\star$ is a simplicial complex.

(3) The 'if' implication is immediate from the $\Gamma$-invariance of $\mathcal{L}$, the connectedness of $\boldsymbol{\Delta}^\star$ and Lemma 2(6). For the converse, we show that if $\mathcal{L}(W) = 0$, then (a) $w_{ij} \in [0,1]$, $(i,j) \in [k] \times [d]$ and $\sum_{i \in [k]} w_{ij} = 1$, for all $j \in [d]$ (and so $W^\Sigma = \mathcal{I}_{1,d}$). The remainder of the proof follows along the same lines as that of [30, Prop. 4.14] except that now for each $j \in [d]$ we have to allow for several rows of $W$ being strictly positive multiples of $v^j$ since $k > d$. $\square$

**Example 3** *Suppose $m = 1$ and $d = 2$. We claim that $\boldsymbol{\Delta}^\star$ is connected. Here the set $\mathfrak{K}^\star$ contains only two partitions: $K = \{\{1,3\},\{2\}\}$, $J = \{\{1\},\{2,3\}\}$. Hence there are two families of matrices in $\boldsymbol{\Delta}(3,2)$*

$$X(\alpha,\beta) = \begin{bmatrix} \alpha & 0 \\ 0 & 1 \\ \beta & 0 \end{bmatrix}, \quad Y(\gamma,\delta) = \begin{bmatrix} 1 & 0 \\ 0 & \gamma \\ 0 & \delta \end{bmatrix},$$

*where $\alpha + \beta = \gamma + \delta = 1$, $\alpha, \beta, \gamma, \delta \geq 0$.*

*Let $\boldsymbol{\Delta}_0^\star$ denote the connected component of $\boldsymbol{\Delta}^\star$ containing $X$ (that is, the arc $X(\alpha,\beta)$, $\alpha + \beta = 1$, $\alpha, \beta \geq 0$). Use the symbol $\sim$ is signify that two families intersect. For example, $X \sim Y$ since $X(1,0) = Y(1,0)$. We claim $(12)^r X \in \boldsymbol{\Delta}_0^\star$. This follows since $X_1 = (13)^r X \sim X$, $X_2 = (12)^r X_1 \sim X_1$, $X_3 = (23)^r X_2 \sim X_2$ and $X_3 = (12)^r X$. It is easy to see that $\boldsymbol{\Delta}^\star$ is isomorphic to a hexagon: 6 vertices, 6 edges and that $\Delta S_2(\boldsymbol{\Delta}(3,2)) = \boldsymbol{\Delta}(3,2)$, where $\Delta S_2 = \Gamma_V$. See Figure 1 where the vertices, connecting edges and symmetries of $\boldsymbol{\Delta}^\star$ are shown.*

*The argument is general and applies when $k > d$—that is, when $V \in M(k,d)$ has at least one row of zeros. The connection can always be made through* row *permutations.*

### Fossilization of critical points, general case.

The phenomenon described above occurs when the network is over-parameterized. In what follows we assume $V \in M(d,d)$ is a matrix with no zero or parallel rows, and extend in the usual way to $V \in M(k,d)$, $k > d$.

Suppose $W \in M(k,d)$ is a critical point of $\mathcal{L}$: in particular, assume that $\mathcal{L}$ is $C^2$ at $W$ and so $W$ has no zero rows. Typically, we assume that $W$ is non-degenerate: all the eigenvalues of the Hessian are non-zero.

Let $\bar{k} > k$ and $\mathfrak{K}^\star$ denote the set of all partitions of $[\bar{k}]$ such that each $\mathcal{K} \in \mathfrak{K}^\star$ has exactly $k$ parts, $K_1, \cdots K_k$ and $j \in K_j$, for all $j \in [k]$.

Just as we did previously, if $\mathcal{K} \in \mathfrak{K}^\star$ and $j \in [k]$, we define the simplex $\Delta_j(\mathcal{K}) \subset \mathbb{R}^{\bar{k}}$, and simplicial set $\Delta(\mathcal{K}) = \prod_{j \in [k]} \Delta_j(\mathcal{K}) \subset M(\bar{k},k)$ of dimension $\bar{k} - k$. Set $\boldsymbol{\Delta}(\bar{k},k) = \cup_{\mathcal{K} \in \mathfrak{K}^\star} \Delta(\mathcal{K})$.

Given $W \in M(k,d)$, $\mathcal{K} \in \mathfrak{K}^\star$ and $\boldsymbol{\delta} \in \Delta(\mathcal{K})$, define $W_\delta \in M(\bar{k},d)$ by

$$\begin{aligned} \boldsymbol{w}_i^\delta &= \delta_{ii}\boldsymbol{w}_i, \ i \in [k] \\ &= \sum_{j \in [k]} \delta_{ij}\boldsymbol{w}_j, \ i > k \end{aligned}$$

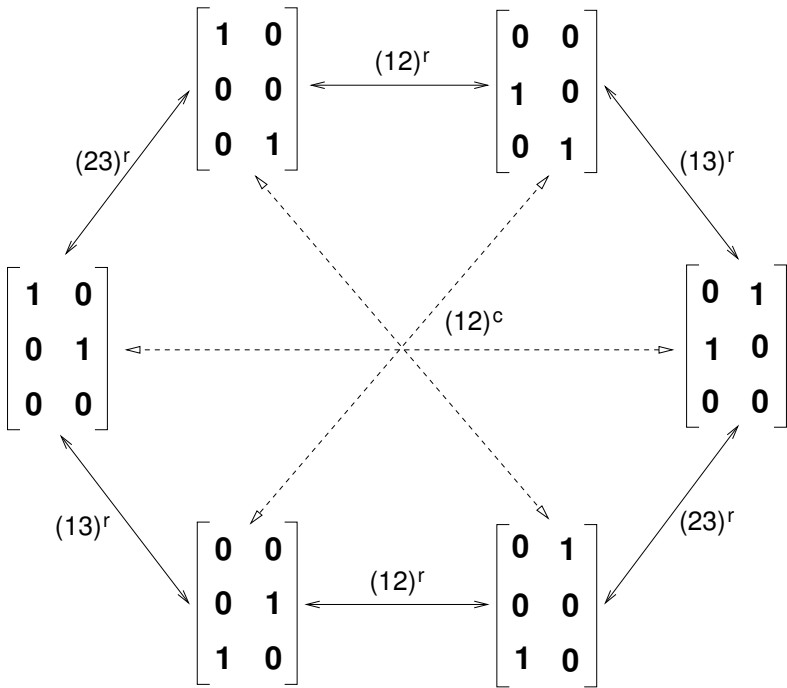

Figure 1: The simplex $\mathbf{\Delta}^\star$ in case $k = 3, d = 2$. Connecting edges are shown using unbroken lines and are labelled by the row transposition that interchanges vertices. The simplicies $\mathbf{\Delta}(3,2), (12)^c\mathbf{\Delta}(3,2)$ are both invariant by $\Delta S_2$.

As we did above when $W = V$, we have $W_\delta^\Sigma = W^\Sigma$. It follows from the definitions that if $\boldsymbol{\delta} \in \Delta(\mathcal{K})$, then $\sum_{\ell \in K_i} \boldsymbol{w}_\ell^\delta = \boldsymbol{w}_i$, for all $i \in [k]$.

Given $W \in M(k,d)$, define $\Delta(\bar{k}, k)(W) = \{W_\delta \mid \boldsymbol{\delta} \in \Delta(\bar{k}, k)\}$ and

$$\mathbf{\Delta}^\star(W) = \Gamma\Delta(\bar{k}, k)(W) \subset M(\bar{k}, d).$$

Set $\mathbf{\Delta}^{\star\star}(W) = \mathbf{\Delta}^\star(W) \smallsetminus \partial\mathbf{\Delta}^\star(W)$

**Proposition 1** *(Notation and assumptions as above.)*

1. $\mathbf{\Delta}^\star(W)$ *is a connected $S_d^r$-invariant simplicial complex of dimension $\bar{k} - k$.*

2. $\mathcal{L}$ *is $C^2$ at all points of $\mathbf{\Delta}^{\star\star}(W)$ and $\nabla\mathcal{L}|\mathbf{\Delta}^{\star\star}(W) \equiv 0$*

3. $\mathcal{L}$ *is constant on $\mathbf{\Delta}^\star(W)$.*

The proof of (1) is similar to that of Theorem 5 (1,2). Since $\mathbf{\Delta}^\star(W)$ is connected, and $\mathcal{L}$ is continuous, (3) follows from (2) ((3) may be proved directly as in Theorem 5 and then (2) follows using the regularity of $\mathcal{L}$ on $\mathbf{\Delta}^{\star\star}(W)$). For completeness, we give a direct proof that $\nabla\mathcal{L}$ vanishes on $\mathbf{\Delta}^{\star\star}(W)$. By an obvious induction, we can reduce to showing that if the partition $\mathcal{K} \in \mathfrak{K}^\star$ satisfies $|K_i| = 1$, $i < k$ and $|K_k| - 1 = n = \bar{k} - k \geq 1$, then $\nabla\mathcal{L}|\Delta(\mathcal{K}) \smallsetminus \partial\Delta(\mathcal{K}) \equiv 0$. Set $m = n + 1 = |K_k|$.

By standard results on $\nabla\mathcal{L}$ [1], $W$ is a critical point of $\mathcal{L}$ if for $i \in [k]$

$$\sum_{j \in [k]} \left( \frac{\|\boldsymbol{w}_j\|\sin(\theta_{\boldsymbol{w}_i, \boldsymbol{w}_j})}{\|\boldsymbol{w}_i\|}\boldsymbol{w}_i - \theta_{\boldsymbol{w}_i, \boldsymbol{w}_j}\boldsymbol{w}_j \right) - \sum_{j \in [d]} \left( \frac{\sin(\theta_{\boldsymbol{w}_i, \boldsymbol{v}_j})}{\|\boldsymbol{w}_i\|}\boldsymbol{w}_i - \theta_{\boldsymbol{w}_i, \boldsymbol{v}_j}\boldsymbol{v}_j \right) + \pi(W - V)^\Sigma = 0$$

$$(15)$$

Suppose that $U \in \Delta(\mathcal{K}) \setminus \partial\Delta(\mathcal{K})$. In order to check whether or not $U$ is a critical point, $\boldsymbol{w}_k$ will be replaced in (15) by $m$ non-zero parallel rows $\boldsymbol{u}_\ell = \alpha_\ell \boldsymbol{w}_k$, where $\sum_{\ell \in [m]} \alpha_\ell = 1$ and $\alpha_\ell > 0$, $\ell \in [m]$. Since the new rows $\boldsymbol{u}_\ell$ are all non-zero and strictly positive multiples of $\boldsymbol{w}_k$,

(A) $\theta_{\boldsymbol{w}_k, \boldsymbol{w}_j} = \theta_{\boldsymbol{u}_\ell, \boldsymbol{w}_j}$, $j \in [k]$, and $\theta_{\boldsymbol{u}_\ell, \boldsymbol{u}_{\ell'}} = 0$ for all $\ell, \ell' \in [n]$.

(B) $\|\boldsymbol{u}_\ell\| = \alpha_\ell \|\boldsymbol{w}_k\|$, $\ell \in [m]$.

(C) $\theta_{\boldsymbol{u}_\ell, \boldsymbol{v}_j} = \theta_{\boldsymbol{w}_k, \boldsymbol{v}_j}$, for all $\ell \in [m], j \in [d]$.

(D) $(W - V)^\Sigma = (U - V)^\Sigma$ (since $\sum_{\ell \in [m]} \alpha_\ell = 1$).

We have $m$ new expressions replacing the right hand side of (15) in case $i = k$:

$$
\sum_{j \in [\bar{k}]} \left( \frac{\|\boldsymbol{w}_j\| \sin(\theta_{\boldsymbol{u}_\ell, \boldsymbol{w}_j})}{\|\boldsymbol{u}_\ell\|} \boldsymbol{u}_\ell - \theta_{\boldsymbol{u}_\ell, \boldsymbol{w}_j} \boldsymbol{w}_j \right) -
$$
$$
\sum_{j \in [\bar{k}]} \left( \frac{\sin(\theta_{\boldsymbol{u}_\ell, \boldsymbol{v}_j})}{\|\boldsymbol{u}_\ell\|} \boldsymbol{u}_\ell - \theta_{\boldsymbol{u}_\ell, \boldsymbol{v}_j} \boldsymbol{v}_j \right) + \pi (U - V)^\Sigma,
$$

where by $\sum_{j \in [\bar{k}]}$, we mean the sum over $\boldsymbol{w}_j$ terms, $j \in [k-1]$, and $\boldsymbol{w}_{\ell'}$ terms, $\ell' \in [m]$. If $\ell \in [m]$, it follows from (A,B) that

$$
\sum_{j \in [\bar{k}]} \left( \frac{\|\boldsymbol{w}_j\| \sin(\theta_{\boldsymbol{u}_\ell, \boldsymbol{w}_j})}{\|\boldsymbol{u}_\ell\|} \boldsymbol{u}_\ell - \theta_{\boldsymbol{u}_\ell, \boldsymbol{w}_j} \boldsymbol{w}_j \right) = \sum_{j \in [d]} \left( \frac{\|\boldsymbol{w}_j\| \sin(\theta_{\boldsymbol{w}_k, \boldsymbol{w}_j})}{\|\boldsymbol{w}_k\|} \boldsymbol{w}_k - \theta_{\boldsymbol{w}_\ell, \boldsymbol{w}_j} \boldsymbol{w}_j \right),
$$

and from (B,C) that

$$
\sum_{j \in [k]} \left( \frac{\sin(\theta_{\boldsymbol{u}_\ell, \boldsymbol{v}_j})}{\|\boldsymbol{u}_\ell\|} \boldsymbol{u}_\ell - \theta_{\boldsymbol{u}_\ell, \boldsymbol{v}_j} \boldsymbol{v}_j \right) = \sum_{j \in [k]} \left( \frac{\sin(\theta_{\boldsymbol{w}_k, \boldsymbol{v}_j})}{\|\boldsymbol{w}_i\|} \boldsymbol{w}_i - \theta_{\boldsymbol{w}_k, \boldsymbol{v}_j} \boldsymbol{v}_j \right)
$$

Noting (D), it follows that $U$ satisfies the critical point equations for $\ell \in [m]$. Along the same lines, but now using the convexity condition $\sum_{\ell \in [m]} \alpha_\ell = 1$, we verify that $U$ satisfies the critical point equations for $i \in [k-1]$. Hence $U$ is a critical point of $\mathcal{L}$. $\square$

# B   Supplementary material for Section 4

We give a brief review of the technique used in this work to compute the Hessian spectrum. The introduction follows [31, 32] verbatim and is provided here for completeness.

Suppose $V \subset \mathbb{R}^m$ is a linear subspace, with Euclidean inner product induced from $\mathbb{R}^m$, and $(V, G)$ is an orthogonal $G$-representation.

**Lemma 3** *[31, Lemma 7, Setion B.1] The representation $(V, G)$ may be written as an orthogonal direct sum $\bigoplus_{i=1}^m (\oplus_{j=1}^{p_i} V_{ij})$ where $V_{ij} \subset V$, $(V_{ij}, G)$ is irreducible, and $(V_{ij}, G)$ is isomorphic to $(V_{\ell k}, G)$ iff $i = \ell$, and $j, k \in [p_i]$. The subspaces $\oplus_{j=1}^{p_i} V_{ij}$ are unique, $i \in [m]$.*

If $p_i = 1$, for all $i \in m$, the orthogonal decomposition given by the lemma is unique, up to order; otherwise the decomposition is not unique. In spite of the lack of uniqueness of Lemma 3, in some cases there may be *natural* choices of invariant subspace for the irreducible components. This is exactly the situation for the isotypic decomposition of $(M(k, k), G)$, $G = S_p \times S_{k-p}$. This naturality allows us to give natural constructions of the matrices $M_i$, $i \in [m]$, used for determining the spectrum of $G$-maps $A : M(k, k) \to M(k, k)$.

The isotypic decomposition for $(M(k, k), S_k)$ is $2\mathfrak{t} + 3\mathfrak{s} + \mathfrak{x} + \mathfrak{y}$, $k \geq 4$ (see Section 4). The subspace of $M(k, k)$ determined by $2\mathfrak{t}$ is the set of all $k \times k$ matrices $\mathcal{T} = \{T_{a,b} \mid a, b \in \mathbb{R}\}$ where the diagonal entries of $T_{a,b}$ all equal $a$ and the off-diagonal entries all equal $b$. There are many ways to write $\mathcal{T}$ as an orthogonal direct sum. For example, $\mathcal{T} = T_{1,1}\mathbb{R} \oplus T_{\frac{2}{k}, -\frac{1}{k(k-1)}}\mathbb{R}$. However, there is only one natural way: $\mathcal{T} = T_{1,0}\mathbb{R} \oplus T_{0,1}\mathbb{R}$. Define $\mathfrak{D}_1^k = T_{1,0}$, $\mathfrak{D}_2^k = T_{0,1}$. If we take the *standard* realization of $(\mathfrak{t}, S_k)$ to be $(\mathbb{R}, S_k)$, where $S_k$ acts trivially on $\mathbb{R}$, then we have natural $S_k$-maps

$\alpha_1, \alpha_2 : \mathbb{R} \to M(k,k)$ defined by $\alpha_i(t) = t\mathfrak{D}_i^k$, $i = 1,2$. If $A : M(k,k) \to M(k,k)$ is an $S_k$-map, then $A$ restricts to the $S_k$-map $A_t : \mathcal{T} \to \mathcal{T}$ and $A_t$ uniquely determines a $2 \times 2$-matrix $[a_{ij}]$ by $A_t(\mathfrak{D}_i^k) = a_{i1}\mathfrak{D}_1^k + a_{i2}\mathfrak{D}_2^k$, $i = 1,2$. The eigenvalues (and multiplicities in this case) of $A_t : \mathcal{T} \to \mathcal{T}$ are the same as the eigenvalues of $[a_{ij}]$. If we choose a different orthogonal decomposition of $\mathcal{T}$, we get a different $2 \times 2$-matrix that is similar to $[a_{ij}]$ and so has the same eigenvalues. The computation of the rest of the eigenvalues follows similarly.

## B.1 Proof of Theorem 3

In Section A.5 algebraic relations between the FPS coefficients were shown to reveal important information on the structure of regular families of critical points. In this section we show how these relations can be further used to evaluate the $\mathfrak{x}$- and the $\mathfrak{y}$-eigenvalues. The method is illustrated for regular families with $b = 4$, $k = d+1$ and isotropy $\Delta(S_{d-1} \times S_1)$.

Referring to notation and results given in Section A.5, any type I or type II family of $\Delta(S_{d-1} \times S_1)$-critical points must satisfy $c_0 \in \{\pm 1\}$ and $c_1 = e_0 = e_1 = e_2 = e_3 = f_0 = f_1 = f_2 = f_3 = 0$. Below we shall assume that $c_0 = -1$. The assumption is not needed, and is only introduced for ease of presentation. For non-zero $g_0$ and $i_0$, the Puiseux series of the eigenvalue associated to the $\mathfrak{x}$-representation is given by:

$$\lambda_{\mathfrak{x}} = -\frac{c_2^2}{2\pi} - \frac{c_2\sqrt{g_0^2}}{2\pi} - \frac{c_2\sqrt{p_0^2}}{2\pi} + \frac{c_4}{2\pi} + \frac{e_4^2}{4\pi} - \frac{1}{\pi} + \frac{1}{4} + \frac{p_2\mathrm{sgn}(p_0)}{2\pi} + \frac{g_2\mathrm{sgn}(g_0)}{2\pi} \qquad (16)$$
$$+ d^{\frac{1}{4}}\left(\frac{c_3}{2\pi} + \frac{p_1\mathrm{sgn}(p_0)}{2\pi} + \frac{g_1\mathrm{sgn}(g_0)}{2\pi}\right) + d^{\frac{1}{2}}\left(\frac{c_2}{2\pi} + \frac{\sqrt{g_0^2}}{2\pi} + \frac{\sqrt{p_0^2}}{2\pi}\right) + O(d^{-\frac{1}{4}}).$$

Thus, the expression $\lambda_{\mathfrak{x}}$ depends on FPS coefficients not determined in Section A.5. We show that $\lambda_{\mathfrak{x}}$ can be evaluated nonetheless, independently of the unknown coefficients.

With FPS coefficients as above,

$$[d^0]F_{d,1} = \frac{c_4}{2\pi} + \frac{e_4^2}{4\pi} + \frac{e_4}{4} + \frac{g_0}{4} + \frac{p_0}{4} - \frac{1}{2\pi} + \frac{p_2\mathrm{sgn}(p_0)}{2\pi} + \frac{g_2\mathrm{sgn}(g_0)}{2\pi},$$
$$[d^{\frac{1}{4}}]F_{d,1} = \frac{c_3}{2\pi} + \frac{p_1\mathrm{sgn}(p_0)}{2\pi} + \frac{g_1\mathrm{sgn}(g_0)}{2\pi},$$
$$[d^{\frac{1}{2}}]F_{d,1} = \frac{c_2}{2\pi} + \frac{\sqrt{g_0^2}}{2\pi} + \frac{\sqrt{p_0^2}}{2\pi}, \qquad (17)$$
$$[d^0]F_{d,2} = \frac{e_4}{4} + \frac{g_0}{4} + \frac{p_0}{4}.$$

Substituting into (16) gives

$$\lambda_{\mathfrak{x}} = \frac{1}{4} - \frac{1}{2\pi} + [d^0]F_{d,1} - c_2[d^{\frac{1}{2}}]F_{d,1} - [d^0]F_{d,2} + [d^{\frac{1}{4}}]F_{d,1}d^{\frac{1}{4}} + [d^{\frac{1}{2}}]F_{d,1}d^{\frac{1}{2}} + O(d^{-\frac{1}{4}}). \quad (18)$$

In particular, $\lambda_{\mathfrak{x}}$ can be expressed in terms of $[d^0]F_{d,1}, [d^{\frac{1}{4}}]F_{d,1}, [d^{\frac{1}{2}}]F_{d,1}$ and $[d^0]F_{d,2}$. Therefore, by continuity, Equation 18 also applies for $g_0 = p_0 = 0$. Since gradient entries vanish at critical points, so do their Puiseux coefficients and so $[d^0]F_{d,1}, [d^{\frac{1}{4}}]F_{d,1}, [d^{\frac{1}{2}}]F_{d,1}$ and $[d^0]F_{d,2}$ vanish, giving $\lambda_{\mathfrak{x}} = \frac{1}{4} - \frac{1}{2\pi} + O(d^{\frac{1}{4}})$. Equation 18 not only gives the exact value of the $\mathfrak{x}$-eigenvalue to $O(d^{\frac{1}{4}})$-order but also describes its sensitivity to variations in the FPS coefficients. For example, it is seen that varying $c_2$ accounts for perturbations of order $O(d^{\frac{1}{2}})$. The derivation of the eigenvalue associated to the $\mathfrak{y}$-representation follows along the same lines, giving

$$\lambda_{\mathfrak{y}} = \frac{1}{4} + \frac{1}{2\pi} + [d^0]F_{d,1} - c_2[d^{\frac{1}{2}}]F_{d,1} - [d^0]F_{d,2} + [d^{\frac{1}{4}}]F_{d,1}d^{\frac{1}{4}} + [d^{\frac{1}{2}}]F_{d,1}d^{\frac{1}{2}} + O(d^{-\frac{1}{4}}). \quad (19)$$

Similar relations exist between *criticality* and *loss*.