# OpenReview forum: "Annihilation of Spurious Minima in Two-Layer ReLU Networks"
_NeurIPS.cc/2022/Conference — NeurIPS 2022 Accept_

### Official Review · Reviewer_XqeF · 2022-07-07

**Rating:** 5
**Confidence:** 2
**Soundness:** 3 good
**Presentation:** 2 fair
**Contribution:** 2 fair

**Summary:**

The paper presents theoretical results on two-layer regression networks with ReLU activations and squared loss. It builds upon a line of work that studies local minima based on symmetries of the loss function (as a function of the weights of the hidden layer). This paper applies and extends previously developed techniques to  the setting where the hidden layer contains more neurons than the input dimension. The new techniques enable the authors to classify local minima in dependence of their symmetry, to provide sharp analytic estimates for the loss at these minima and the spectrum of the loss Hessian, and to thereby understand when local minima of certain symmetries turn into saddles when more neurons are added to the network.

**Questions:**

- Why is it reasonable to study permutations of the input neurons?

- Is it possible to provide specific weight matrices that cover the different symmetry classes?

- Are all spurious local minima (possible under the given setting) covered by the theory?

**Limitations:**

The paper makes strong assumptions. It is unclear whether the local minima can be found in practical neural networks. The necessity of some strong assumptions and their consequences are not discussed in detail

**Strengths And Weaknesses:**

Strengths:

- The paper applies interesting and novel techniques by combining a variety of mathematical tools.  It builds upon a recent line of work, but applies the approach to a novel setting (where the hidden layer contains more neurons than input dimensions) and thereby requires new ideas and techniques. This results in novel, analytical results that provide sharp estimates on the loss at  certain local minima and the spectrum of the loss Hessian. Interesting results can be then deduced on how minima turn into saddles in dependence of their symmetry class. (Some minima turn into saddles when a single neuron is added, while others require more neurons.) The analytic results do not only provide existence of certain families of local minima, their loss and Hessian, but also their frequency of appearance.

- While the paper assumes knowledge of the line of papers it builds upon, all theoretical results are proved and detailed calculations are appended in the supplements.


Weaknesses:

- The paper makes a number of assumptions, which questions the applicability of the results to networks in practice:
The authors consider only two-layer regression networks with squared loss in a teacher-student setting, where labels are generated by a target network. The target network is fixed to a weight matrix equal to an identity matrix (stacked with zeros if the hidden layer contains more neurons than input dimensions) and the weights of the output layer equals a vector of ones. (The paper claims that an extension to trainable output layers is possible, but no details or further explanation is given. Similar, the authors write that , while the results assume a target of high symmetry, this would not be necessary, but again without evidence or further discussion.) This restricted setting can be slightly generalized to a hidden weight matrix given by an orthogonal matrix of size input dimension and rest of zeros, but only because of another strong (unrealistic) assumption of a Gaussian input.

- It is the latter assumption that seems crucial to the entire theory, while also implying that the setting is quite different to a realistic setting. In particular, local minima are classified into groups of symmetry, defined by being invariant under permutations of hidden and input neurons. In particular, the authors want the target network to be invariant under permutations of the input neurons, which seems to require strong assumptions on the input distribution. (Aside, I would appreciate a clarification from the authors why it would be interesting to consider permutations of the dimensions in the input space at all.)

- A second perceived weakness is the lack of clarification how exhaustive the results are. Are all spurious local minima (possible under the given setting) covered by the theory? It seemed to me that the answer is no, and the paper only considers very specific families of critical points.

- The paper should also be better placed in related work. Existence of spurious local minima in networks have been shown previously (using other techniques) and it remains unclear, how the results of this paper relate to these.  To give a possibly, strongly related example, a classical paper by Fukumizu and Amari („Local minima and plateaus in hierarchical structures of multilayer perceptrons Neural Networks“ 13, 2000) provides specific weight combinations (with a certain symmetry) that lead to local minima. To better place the given submission into the related work, the authors could discuss the specific form of weight matrices that are covered by the symmetry classes. (Since the group actions are very specific, they translate to specific weight configuration, which could be made explicit.) Related to this: If one were to translate the results into statements on specific weight configurations in a network, which certainly seems possible, are the techniques to study the local minima necessary, or can the results be interpreted more easily?

-------
Additional Comments:

I appreciated the detailed calculations in the appendix and detailed arguments in some parts. However, I also found that the paper too often assumes implicit knowledge and is not too precise in its use of terminology. I append a list of some of these points, where the presentation could be improved. Despite strong efforts, I was therefore not able to follow all of the arguments in the paper (I did look into the line of work this submission builds upon, but did not study it in detail.)

The authors call the their setting as “overparameterized“, when the hidden layer has more neurons than the number of input dimensions. This is in conflict with existing works that call a network overparameterized when it has more parameters than input patterns to learn.

Line 116  “The loss function is always invariant...“ This only holds if the same permutation as applied to the output weights

Line 120, the meaning of the equality sign is not defined

Line 126: “L is invariant“. I suppose, the authors mean that the teacher network is invariant.

Line 134: what means “fixed“? V was a fixed value to start with. Probably, the authors mean “invariant under“?

Line 138: what exactly is meant by the terminology “nondegenerate“ and “spurious“?

Line 138,139 is missing a reference for the empirical evidence

Line 143, reference is missing to the “recent work“

Line 163, The notation G_c(d) is not defined

Line 185: Def 2 requires more explanation “corresponding to the action of delta S“,
Possibly it should also be discussed in more detail why the diagonal elements converge to either +1 or -1.

Line 198 “Since the loss at type II minima decays...“ Please provide an explanation or a  reference

Line 267 Typo: Adding one neuron should change the second subscript

Line270: “This isotypic component may be written as...“ Please provide an explanation or a  reference

Line 357: This second part of conclusion is lacking context to understand this part. In particular, please revise or explain the following terminology:
“sink“, “source“, “index“, “local deformation of the landscape geometry“, “bifurcation theory“, “forced symmetry breaking leads to great complexity near the transition but minimal models of complexity can be given“

---

> ### Author Response · Authors · 2022-07-30
> **Response to Reviewer XqeF**
>
> Typos\suggestions, including those not listed below, have all been adopted.
>
>
> **Assumptions and scope of applicability.**
>
> Existing theoretical techniques aimed at addressing ReLU networks in complete generality fail to rigorously explain why optimization is possible in deep learning. This is a barrier inherent to distribution-free approaches (Blum \& Rivest (1992);  Brutzkus \& Globerson (2017); Shamir (2018)). We approach the problem from a standpoint taken by [10, 12, 34, 27, 20, 36, 5, 30]. That is, rather than striving for generalization, our emphasis is on a rich model that helps elucidate some of the key foundational questions in a framework amenable to detailed and rigorous analysis. Orthogonality of the target matrices is not required by the symmetry-breaking framework: other choices of target matrices, distributions, activation functions and architectures have been addressed in previous works, and are a topic of our current research. As mentioned briefly in the concluding comments and in this response, when $d$ is fixed and the symmetry of $V$ is broken the phenomena we describe persist. Of course, for a general asymmetric $V$ there is not much quantitative one can say about spurious minima. Our expectation is that the analysis of symmetric case and symmetry breaking perturbations [cf. [6]) will help inform and guide the formulation of the right questions in the general case. For example, the existence of different types of spurious minima with different asymptotics of the decay of  the loss, or spectral properties of the Hessian (see [4,5] for more on the latter point).
>
>
> > ... Why it would be interesting to consider permutations of the dimensions in the input space at all.
>
> Invariance properties of the input space are widely-believed to be one of the key factors for the success of deep learning (e.g., the influential paper ‘Convolutional networks and applications in vision’ by LeCun et al.). Any symmetry group related to the input space can be embedded in a permutation group (Cayley's theorem). In our setting, the invariance to subgroups of the permutation group considered in the paper is reflected in the structure of critical points, making possible the use of powerful methods for analytically characterizing the loss landscape.
>
>
> > Are all spurious local minima (possible under the given setting) covered by the theory?
>
> Most likely not - if by all you mean all isotropy types. Certainly, we do not expect the case where there is no symmetry or very small symmetry groups. For fixed $d$, when the symmetry of $V$ is broken the minima will all persist (assuming the Hessians all have non-zero eigenvalues) but there will be no symmetry. Please see also general comment 4 above.
>
>
> > Existence.. have been shown previously.
>
> Our goal in this work was not to formally establish the existence of minima, but rather to study the mechanism which makes the loss landscape accessible to simple optimization methods — despite being highly nonconvex. To this end, we use the rich symmetry structure to characterize the Hessian spectrum at different families of critical points and investigate how increasing the number of model parameters (i.e., hidden neurons) turn spurious minima into saddles. Our results are also precise - for example, exact spectrum up to given order in $1/d$.
>
> > Translate the results into statements on specific weight configurations.. ?
>
> The structure of weight matrices for the class of isotropy groups considered in the paper is well-understood. For example, matrices of isotropy $\Delta S_d$ lie in the two-dimensional space {$ a I_d + b I_d I_d^\top |  a,b \in \mathbb{R}$}. A detailed study is given in [4, 7].
>
>
> > “overparameterized“... is in conflict with existing works
>
> Terminology in our field has yet to stabilize. Over-parameterization, as an indication for richer student models, has been used for example in (but certainly not only) https://arxiv.org/pdf/1901.09085.pdf and https://arxiv.org/abs/1712.08968. Other (plenty of) examples exist. We will add a remark clarifying the possible naming collision.
>
> > Line 116 only holds if the same permutation as applied to the output weights
>
> Please see general comment 3 above.
>
> > Line 120, the meaning of the equality sign is not defined
>
> The sign $\approx$ indicates a group isomorphism.
>
>
> > Line 126:.. I suppose, the authors mean that the teacher network is invariant.
>
> Note, the teacher network is *not* $S_k \times S_d$-invariant. Rather, the loss function *is* $S_k \times S_d$-invariant. It is perhaps somewhat surprising at first glance.
>
>
> > Line 138: what exactly is meant by the terminology “nondegenerate“ and “spurious“?
>
> ‘Spurious’ (line 18) is a widely used to indicate a non-global minimum (E.g., https://arxiv.org/abs/1712.08968 and https://papers.nips.cc/paper/2016/file/7fb8ceb3bd59c7956b1df66729296a4c-Paper.pdf)
>
> ‘Nondegenerate’ (line 133): “non-degenerate critical points (no zero eigenvalues)”
>
>
> > Questions:
>
> Please see the answers above.

---

> > ### Comment · Reviewer_XqeF · 2022-08-07
> > **thank you for the detailed response**
> >
> > I thank the reviewers for their detailed reply. A few points that may be worth discussing:
> >
> > I do not understand the clarification in the main comment 3. The activation function is only positive homogeneous, so the second layer can only be reduced to a vector of (plus) ones (using homogeneity) if all weights are positive.
> >
> > I would argue that it is important to distinguish between invariance to irrelevant variations of the input with a physical meaning (similar to the translation-equivariance of convolutional layers), and symmetries of the loss function based on permutations of input neurons. Am I missing an intuitive understanding of these type of symmetries?
> >
> > I understand that the paper does not aim to show existence of non-global local minima, but it could nonetheless be placed into related work on non-global minima and generalized saddle points. If the (isotropy groups of) local minima are well-understood, then they can be related to other local optima found in related work.
> >
> > One minor point aside, I don't think the presentation is hard to follow at some parts because the reviewers not trained in algebraic geometry and representation theory, but that the paper assumes knowledge of those research paper which the current submission builds upon.

---

> > > ### Author Response · Authors · 2022-08-09
> > > **Response to reviewer XqeF**
> > >
> > > > I thank the authors for their detailed reply. A few points that may be worth discussing:
> > >
> > > Thank you for your questions and remarks. They have given us an opportunity to better understand how different parts of our paper are perceived, and to hopefully improve the presentation of our work.
> > >
> > > All points below shall be clarified in the text.
> > >
> > > &nbsp;
> > >
> > > > I do not understand the clarification in the main comment 3. The activation function is only positive homogeneous, so the second layer can only be reduced to a vector of (plus) ones (using homogeneity) if all weights are positive.
> > >
> > > For the concrete families of minima investigated in [6], the weights of the second layer are positive, and can be therefore rescaled to plus ones. Critical points with negative weights in the second-layer are not covered in [6]. (Indeed, more and more critical points with different structure and isotropy are being discovered and classified as investigation progresses, see also general comment 4 above.) All families of minima studied in the present work have positive second-layer weights, and are therefore amenable to the same (straightforward) reduction used in [6].
> > >
> > > &nbsp;
> > >
> > > > I would argue that it is important to distinguish between invariance to irrelevant variations of the input with a physical meaning (similar to the translation-equivariance of convolutional layers), and symmetries of the loss function based on permutations of input neurons. Am I missing an intuitive understanding of these type of symmetries?
> > >
> > > At this time, it is not clear whether a conclusive answer to your question exists. Indeed, the results shown in the present work can be interpreted as suggestive evidence in favor of a different explanation, namely: it is the very intricate interplay between symmetries that are inherent to data distributions and those that are (“forcibly”) incorporated into neural network models that make such blends successful.
> > >
> > > Concretely, in our setting:
> > > 1. Invariance properties of the loss function are determined by symmetries of the neural network model and the underlying distribution.
> > > 2. These properties are reflected in the symmetry (isotropy) of spurious minima.
> > > 3. Since increasing the number of student neurons does not affect the isotropy type, spurious minima must, provably, transform into saddles.
> > >
> > > &nbsp;
> > >
> > > > I understand that the paper does not aim to show existence of non-global local minima, but it could nonetheless be placed into related work on non-global minima and generalized saddle points. If the (isotropy groups of) local minima are well-understood, then they can be related to other local optima found in related work.
> > >
> > > Families of minima of the loss landscape considered in the present work are classified by their isotropy type and their type (the latter being determined by the sign of the diagonal entries). This taxonomy has been kept largely consistent throughout the line of work on symmetry-breaking. Therefore, in all existing works, ``type II $\Delta S_{d-1}$-minima’’ indicates the same family of minima – In particular, minima relate to one another across *different* works by their very classification.
> > >
> > > The loss landscape exhibits a "zoo" of minima and critical points which, owing to symmetry, can be nonetheless organized in a systematic way (just described). This zoo has yet to be completely understood and characterized, especially in the case when $k > d$. For example, there can exist distinct families of critical points with the same limiting behaviour but different Hessian spectrum (e.g. a family of minima and a family of saddles). Further, the path-based techniques on the fixed point space allow us to identify regular families with isotropy $\Delta (S_q \times S_q)$ - here $d = 2q$ will be even and, for fixed $q$, the critical point with isotropy $\Delta( S_q \times S_q)$ can appear in a family $\Delta(S_{d-q} \times S_q)$ where now $q$ is fixed and $d$ is varied.
> > >
> > > In contrast, the structure of matrices belonging to a given isotropy subgroup *is well-understood*. For example, any matrix of isotropy $\Delta S_d$ is necessarily a linear combination of the identity and the all-ones matrix. See [4, 7] for a complete account.
> > >
> > > We are happy to elaborate more on this point if we misunderstood your question.
> > >
> > > &nbsp;
> > >
> > > > One minor point aside, I don't think the presentation is hard to follow at some parts because the reviewers not trained in algebraic geometry and representation theory, but that the paper assumes knowledge of those research paper which the current submission builds upon.
> > >
> > > It is true that summarizing the “prerequisites” has caused us some problems on account of the required background (representation theory, FPS techniques, and path based methods - giving dependence on real parameter d). As well as working on the preliminaries in the main paper, we will make the supplementary material broader in scope so as to make the submission more self-contained.

---

> > > > ### Comment · Reviewer_XqeF · 2022-08-09
> > > > **response to the authors**
> > > >
> > > > I again thank the authors (! this time correctly addressed) for their clarifying reply.
> > > >
> > > > If I understand correctly, the generalization to trainable second layers only applies if all second layer converge duuring training to positive weights, a limitation that I think should be mentioned. Secondly, the necessary theoretical assumptions on the symmetries in the underlying input distribution still seem rather strong to me. The example of (isotropy groups of) local minima under consideration reflect (and require) the symmetries in the input distribution. The theoretical setting is therefore different to neural networks trained on finite data and it may be difficult to relate the findings to local minima and saddle points in the more practical setting; a relation I was hoping for when asking for a comparison to related work.
> > > >
> > > > As a result, my evaluation of the paper remains unchanged. As a theoretical paper, the submission certainly offers novel techniques to prove novel, theoretical results. However, the applicability of the theory seems to require strong theoretical assumptions, and it remains unclear to me how the techniques and results can be  carried over to a more realistic setting. Taken together, my evaluation remains that this is a "technically solid paper where reasons to accept outweigh reasons to reject".
> > > >
> > > > I appreciate the author's intent to extend and polish the presentation to make the paper more self-contained, which can contribute to the understanding of readers not sufficiently familiar with previous work on symmetry-breaking.

---

> > > > > ### Author Response · Authors · 2022-08-09
> > > > > **Response to reviewer XqeF**
> > > > >
> > > > > Thanks for the detailed responses.
> > > > >
> > > > > &nbsp;
> > > > >
> > > > > Statistical methods, often from theoretical physics, have so far not been particularly successful in explaining phenomena seen in neural networks. For example, explaining how adding neurons can remove spurious minima---the main focus of our paper. Our approach is to start from an analytically tractable case---one already acknowledged as difficult: ``As far as we know, an analytical expression for the roots might not even exist'' (Safran & Shamir, [Introduction, 29]).
> > > > >
> > > > > &nbsp;
> > > > >
> > > > > As indicated in the concluding comments, the results proved hold under symmetry breaking perturbations:
> > > > > *we are describing phenomena that are robust and so already our results have the power to disprove or support general conjectures in the field* (for example, on Hessian spectra [5,6]). At this point, we feel it is premature to attempt a `general theory' since our focus is still partly exploratory though guided by properties of the symmetric models that can be expected to hold in more general situations where symmetry cannot be invoked.
> > > > >
> > > > > &nbsp;
> > > > >
> > > > > At this stage, we want to try and keep things relatively simple but certainly agree with the need to expand the scope to include neural networks trained on finite data. Following a recent series of works [4, 26, 35, 29] and Hardt et al., 2016; Hardt & Ma, 2016, we focus on the generalization error. Various properties of the training loss can be readily deduced by concentration of measure arguments, e.g., using generalized vector-valued Rademacher complexity (e.g., Foster et al. 2018, Mei et al. 2017).

---

### Official Review · Reviewer_WAgA · 2022-07-08

**Rating:** 6
**Confidence:** 3
**Soundness:** 3 good
**Presentation:** 2 fair
**Contribution:** 4 excellent

**Summary:**

This paper focuses the setting of fitting a two-layer networks to a target network with respect to the square loss. They apply novel tools from algebraic geometry to utilize the symmetric structure and characterize how over-parameterization annihilates spurious local minima.


**Questions:**

Can the results in the paper be extended to the empirical loss, i.e., the loss function is only evaluated at limited number of data points?

Let $N$ be the number of data points. It is assumed that the labels are generated by a target network with $d$ neurons. Is it possible to generalize the results to a target network with more neurons, say $N+1$? From the Caratheodory’s theorem, an optimal neural network fitting arbitrarily label needs at most $N+1$ neurons. Thus, the results will be more influential if it can be generalized to arbitrary target network.


**Limitations:**

Yes.

**Strengths And Weaknesses:**

Strength:
- This is a strong theoretical paper discussing the spurious local minima of two-layer ReLU networks.
- The tools from algebraic geometry and representation theory are novel for the study of local minima in two-layer neural network. The analysis of the families of local minima and Hessian spectrums can bring meaningful results for the theoretical study of two-layer networks.

Weakness:
- The paper should be self-contained. However, the notation is confusing and this imposes difficulty for the reader to understand the contribution of the paper.
    -  In Line 122, it is said $V=I_d\in M(k,d)$ but here V is a d-by-d matrix and we need to append zero rows to V. It will be better to use another notation.
    - In Line 152, what does $i_g:\Delta S_d\subset \Delta S_{d+1}$ mean?
    - Some notations are not defined. For instance, what is $\mathcal{L}|M(k,d)^G$ in line 150 and what is $\mathcal{L}|\mathbb{R}^n$ in line 165?
- The optimization variable is conflicted. (W,\alpha) is the pair of optimization variable for the loss $\mathcal{L}$ and a critical point should be in $M(k,d)\times \mathbb{R}^k$. In Line 150, why $c \in M(k,d)$ is a critical point? It seems that the results of the entire paper focuses on optimizing W alone. Please specify what is the optimization variable.
- The paper states that the over-parameterization annihilates spurious local minima but the main theorems (Theorem 2-4) only discuss the cases when $k=d,d+1,d+2$, which is not quite related to the regime of over-parameterization.

---

> ### Author Response · Authors · 2022-07-30
> **Response to Reviewer WAgA**
>
> Typos\suggestions, including those not listed below, have all been adopted.
>
> > Presentation and background.
>
> Please see general comments 1 above.
>
> > In Line 152, what does $i_g:\Delta S_d \subset \Delta S_{d+1}$ mean?
>
> Please see general comments 2 above.
>
> > Some notations are not defined. For instance, what is  in line 150 and what is  in line 165?
>
> The symbol $F|_X$ is conventionally used to denote the restriction of the domain of $F$ to the set $X$. Fixed point spaces $M(k,d)^G$ are defined in Section 3.
>
> Please let us know if this addresses your concerns properly.  We have also made changes in notation in the supplementary materials.
>
> > The optimization variable is conflicted... the optimization variable.
>
> Since the weights of the second layer are assumed to be normalized to one (allowed by homogeneity), the families of minima we study can be described only by the weights corresponding to the first layer. Our goal is then to investigate how the extremal properties of a given family of minima varies when the number of student neurons is increased. We found that adding neurons turns minima into saddles, thus indicating why gradient-based methods might succeed in detecting minima with an improved (lower) loss. Please also see general comment 3 above.
>
> > The paper states that the over-parameterization annihilates spurious local minima... the regime of over-parameterization.
>
> Quite remarkably, already adding one and two neurons affects the loss landscape in a dramatic way, transforming certain families of minima into saddles. In fact, it is unclear whether bad local minima exist at all if more than two neurons are added. This has been also observed by [29]. Our work is the first to provide a rigorous explanation of this peculiar phenomenon for minima of certain symmetry, and indicate why this might be the case for all minima. We discuss the case where $k-d>2$ in more detail in general comment 5 above.
>
>
> > Can the results in the paper be extended to the empirical loss, i.e., the loss function is only evaluated at limited number of data points?
>
> That is a very interesting question: current theoretical bounds for the number of samples required for the training loss to uniformly converge (UC) to the population loss are worst-case and are known to become completely vacuous when applied to parameter regimes encountered in practice (cf., [https://arxiv.org/abs/1703.11008]). We hope that the symmetry-based approach might provide a different perspective (not based on UC) as to why SGD succeeds in these highly non-convex problems nonetheless. Preliminary numerical experiments we conducted confirm that minima of the training loss are also (approximately) highly symmetric. Thus, potentially, rather than worst-case notions of sample complexity measured over the space of all possible weight matrices, improved generalization bounds may be obtained by focusing on a restricted set of highly symmetric matrices.

---

> > ### Comment · Reviewer_WAgA · 2022-08-04
> > **Response to the authors**
> >
> > The authors address most of my concerns and I raise the score.

---

### Official Review · Reviewer_DYe8 · 2022-07-10

**Rating:** 7
**Confidence:** 1
**Soundness:** 3 good
**Presentation:** 3 good
**Contribution:** 3 good

**Summary:**

This paper concerns about the structure of critical points and how local minima change to saddles in over-parameterized regime in the case of 2 layer ReLU networks, square loss, and labels created by planted model. Using symmetry structure of loss function and considering certain symmetric types of critical points it is shown that adding neurons can turn non-global minima into saddles, addressing over-parameterized models.

**Questions:**

Typo:

lines 599 and 686, table 5.2 should be replaced by table 2. line 527, $M(d+2,d)^{\Delta S_{d-1}}$. line 538, $\Omega_1$. line 606, restricted "to" the. line 619, $M(d+2,d)$. line 250, by section A.7 do you mean section B.1 where you prove Theorem 3?


Questions:

I can't understand why the loss function is in invariant by the group of row permutations (line 116). shouldn't the weights of 2nd layer play a role here? why do you just consider $W$? I think I'm convinced by looking at line 124 where you set 2nd layer to be all ones, but this should come before line 116 to avoid confusion.

What will be the main barrier in using these methods for analyzing k-d>2?

For line 198, where do you show that the loss at type II minima decays as $\Theta(\frac{1}{d})$? and is $\Theta(1)$ loss at type I minima characterized by equations in lines 210-211?

Regarding Definition 2, are there only two types of critical points for all k, d or just the cases you consider? I can see in Theorem 2 you prove there are 1 or 2 family of critical points for specific isotropies. Also for the remarks in lines 337-340, is there any evidence that the 2 families mentioned are representative of other symmetric spurious minima?

for line 200, why it is reasonable for the loss at initialization to be compared to loss in minima?

Regarding line 347, can you point out where in the paper/proofs you assumed V having high symmetry? is it the assumption of $V=I_d$ in line 126?

In section B1, what is an eigenvalue transition matrix?

What is the meaning of subscripts 1,2,3 in $\nabla \mathcal{L}$ in beginning of page 19?



**Limitations:**

I don't see any negative societal impact, except for misuse\illegal use of Neural networks.

**Strengths And Weaknesses:**

Assuming the reader has enough knowledge of existing work and tools being used, the paper is well-written and addresses the effect of over-parameterization on certain symmetry types of minima and their transformation to saddles. The FPS method for defining spectrum of Hessian looks powerful for analyzing the loss and Hessian at local minima, and Theorem 3 is a good example of Hessian spectrum characterization at certain family of critical points. I like the arguments where adding neurons changes the nature of critical points, like in lines 280-284 and 601-603, as well as Theorem 4, and how different families of minima require different number of additional neurons to turn into saddles. I couldn't digest most of the proofs, except for pages 19-21, as I am not familiar with this line of research revolving around representation theory.

The paper also assumes previous knowledge from previous works and is not highly self-readable; However authors try to familiarize readers with existing work in introduction section.
for example, regarding lines 72-76, authors indicate a previous work and pinpoint that considering symmetry type of minima is important.Also following line 77, the symmetry breaking phenomenon observed in previous works is outlined.

---

> ### Author Response · Authors · 2022-07-30
> **Response to Reviewer DYe8**
>
> Typos\suggestions, including those not listed below, have all been adopted.
>
> > I can't understand why the loss function is in invariant by the group of row permutations (line 116). shouldn't the weights of 2nd layer play a role here? why do you just consider $W$? I think I'm convinced by looking at line 124 where you set 2nd layer to be all ones, but this should come before line 116 to avoid confusion.
>
> Please see general comment 3 above. If all weights of the second layer are set to one, the network output is simply the sum of the values of the hidden neurons. Summing the values in a different order (i.e. permuting the rows) does not affect the output. That’s the easy part. The second part of the invariance follows from the properties of the underlying data distribution and the structure of the teacher matrix. See [4] or [7] for a direct derivation.
>
>
> > What will be the main barrier in using these methods for analyzing $k-d>2$?
>
> Please see general comment 5 above.
>
> > For line 198, where do you show that the loss at type II minima decays as $\Theta(\frac{1}{d})$? and is $\Theta(1)$ loss at type I minima characterized by equations in lines 210-211?
>
> For $k=d$, this is established in [5] and [6] and uses the fractional power series (FPS) expansion substituted into the loss (only initial terms of the FPS are important in determining initial terms of the FPS of the loss at a minimum). For $k = d+1$, this is given in Figure 1 and indeed lines 210-211.
>
>
> > Regarding Definition 2, are there only two types of critical points for all k, d or just the cases you consider? I can see in Theorem 2 you prove there are 1 or 2 family of critical points for specific isotropies. Also for the remarks in lines 337-340, is there any evidence that the 2 families mentioned are representative of other symmetric spurious minima?
>
> For the isotropy types considered in the paper, the only types that occur are type I and type II. Empirical evidence suggests that this applies rather generally. In the extensive numerical experiments we have conducted minima (and critical points which are minima for the loss restricted to the fixed point space) were always type I or type II, regardless of their isotropy type.
>
> > for line 200, why it is reasonable for the loss at initialization to be compared to loss in minima?
>
> While type II minima are detected by SGD, type I are not. Since the loss at type II, $\Theta(\frac{1}{d})$, is smaller than the loss at type I, $\Theta(1)$, one might argue that the latter is not detectable by SGD since the (expected) loss at initialization is $\ll \Theta(1)$. However, this turn out not to be the case as we prove that the initial loss is $\Theta(d)$, which is larger than the loss at both types of minima.
>
>
> > Regarding line 347, can you point out where in the paper/proofs you assumed $V$ having high symmetry? is it the assumption of $V=I_d$ in line 126?
>
> Indeed, that has been clarified.
>
> > In section B1, what is an eigenvalue transition matrix?
>
> For each copy of a given irreducible representation, a representative vector has been chosen. As the Hessian matrix is stable on isotypic components, its action on representative vectors belonging to the same irreducible representation may be described in terms of a matrix, namely the transition matrix. A complete account of the technique is given in section 3 “The method: a symmetry-based analysis of the Hessian” in [5].
>
> > What is the meaning of subscripts 1,2,3 in $\nabla \mathcal{L}$ in beginning of page 19?
>
> The subscript denotes the entries of the gradient map. We have worked to make the notation clearer throughout the paper. In particular, A1 - A6 are completely rewritten.

---

> > ### Comment · Reviewer_DYe8 · 2022-08-08
> > **Response**
> >
> > I read all comments and other reviews and will keep my score as well as confidence fixed!

---

### Official Review · Reviewer_fCuP · 2022-07-10

**Rating:** 5
**Confidence:** 3
**Soundness:** 3 good
**Presentation:** 1 poor
**Contribution:** 2 fair

**Summary:**

The authors extend the theory of [6] for mild overparameterization where the student has one or two more neurons than the teacher (ReLU activation function).
For several chosen families of local minima, the authors give an analytic description in terms of the loss at these minima and their Hessians. The neuron additions can turn minima into saddles, "minima of lesser symmetry" needs more neurons for turning them into saddles.

**Questions:**

Is there any local minima of type $\Delta S_{d-p} \times \Delta S_{p}$ where $p>2$? As far as I see, only $p=\{0,1,2\}$ are discussed (also, some overparameterization).

line125: I don't think it is trivial at all whether the two-layers setup can be studied in the same way as the one-layer setup studied in the paper. Can the authors please comment on this? As far as I see, also in the reference [6], only the one-layer case is studied.

Typos/detailed comments:
line 60-61: adding one neuron results in these minima ... unclear sentence
line 101: citation needed
caption of Table1: I don't understand how lines 4-6 and lines 10-12 are related to the table.
line 116: $\mathcal{L}(\sigma W, \sigma \alpha) = \mathcal{L}(W, \alpha)$ for two-layers networks. However, the authors only study the committee machine setup where the second layer is fixed at $1$. Why not introduce the setup already in this form in page 1 (Eqs~1 and 2)?
line 152: definition of inclusion is unclear to me
line 289: (typo) so turn minima -> to turn minima
line 367: what is meant by non-integer values of $d$?



**Limitations:**

The paper is written in a neutral way, and the written text reflects the findings in the theorems.

**Strengths And Weaknesses:**

Originality: The paper extends the theory of [6]. Although the existence of local minima in the asymptotic regime and characterization in [6] is exciting as an example neural network landscape with provable local minima families, I find the contributions of the current paper somehow incomplete, therefore less attractive.

Quality: The analysis techniques are non-standard. I did not have time to check the proofs, but the results look rigorous. No experiments are presented in the main text, for example, to justify the claim of why type-II minima are more attractive.

Clarity: Although individual sentences usually can be understood, the technical terms used in this paper are hard to follow. Table I is incomprehensible, the caption refers to the material in the later sections, instead of explaining the table. The spectrum column cannot possibly represent the full Hessian spectrum since there is a scalar value, does it refer to the min. eigenvalue of the Hessian? What are the r and n representations?

Significance: I think it is an important problem to classify all the critical points of the neural network landscape in the paper. This analysis can pave the way to explaining the difficulties in training mildly overparameterized networks. However, the paper only classifies a few types of local minima points in the asymptotical limit, and it is not even clear whether the dynamics would converge to these points or not.

---

> ### Author Response · Authors · 2022-07-30
> **Response to Reviewer fCuP**
>
> Typos\suggestions, including those not listed below, have all been adopted.
>
> > Identifying all minima.
>
> Please see general comment 4.
>
> > line125: I don't think it is trivial at all whether the two-layers setup can be studied in the same way as the one-layer setup studied in the paper. Can the authors please comment on this? As far as I see, also in the reference [6], only the one-layer case is studied.
>
> Please see general comment 3.
>
> > line 60-61: adding one neuron results in these minima ... unclear sentence
>
> Adding one hidden neuron to the student network. I.e., k = d + 1.
>
> > line 152: definition of inclusion is unclear to me line
>
> Please see general comment 2.
>
> > line 367: what is meant by non-integer values of $d$?
>
> The symmetry-based technique used in the paper yields a family of minima which depends continuously on the dimension parameter d. These minima, and the associated detailed analysis, lie in a fixed point space of dimension independent of $d$. Integer values of $d$ correspond to critical points lying in $M(k, d)$. However, $d$ may also assume non-integer (that is, fractional) values for which the spectrum analysis is still applicable - this makes essential use of symmetry and representation theory. For example, the determinant of the Hessian is well defined as a function of real $d$ for these families!  In particular, we found that certain families of critical points change (i.e., *bifurcate*) from saddles to minima at non-integer values of $d$.

---

### Author Response · Authors · 2022-07-30
**General comments**

We thank the reviewers for their time, detailed feedback and helpful critical comments.

1. **Presentation and background**. The mathematical tools used in the work are currently not part of standard training programs of researchers in machine learning. Indeed, it is only recently that certain tools from algebraic geometry and representation theory were found to be effective in the context of the theory of deep learning. Balancing between the presentation of our results on annihilation of minima under overparameterization, a decent account of relevant background material and the symmetry-based technique developed in the recent line of work, is therefore a subtle task. We put a lot of thought into how to present the new methods and appreciate the generally positive response of the reviewers to our efforts. The reviews have been helpful in identifying some points which  require further elaboration. We will revise accordingly.

&nbsp;

2. **Inclusions**. Formally speaking, $\Delta S_{d+1}$ does not contain $\Delta S_d$ as a subgroup, but rather isomorphic copies thereof. For example, the group $S_4$ can act on a set of five elements by leaving (say) the last element fixed and permuting the first four elements, thus embedding $S_4$ in $S_5$. Likewise, inclusions are injective group homomorphisms which embed $\Delta S_d$ in $\Delta S_{d+1}$. We will make this clearer in the text.

&nbsp;

3. **Trainable vs non-trainable second-layer**. Reference [6] directly addresses the case where the second-layer is trainable (see Theorem 1). For families of minima investigated in [6], the homogeneity of the network allows one to carry out the analysis by first reducing to the case where the weights of the second layer are normalized to one, and then rescaling. The loss landscape considered in the paper exhibits symmetry breaking minima and is therefore amenable to the same methods. We felt that pursuing this generalization would reduce readability, especially in view of the use of novel mathematical techniques. We shall clarify this point in the paper.

&nbsp;

4. **Identifying all minima**. Families of minima exist of isotropy $\Delta S_{d-p} \times S_p$ for $p>2$. All the families of minima we have encountered so far have large symmetry groups, typically maximal proper subgroups of $\Delta S_d$, and are amenable to the same symmetry-based analysis used in the paper. While we cannot yet rule out the existence of families of minima with no symmetry (trivial isotropy), empirical work shows their  existence is unlikely even for large values of d.  The use of symmetry enables the description and detailed analysis of many families of minima and critical points in a problem which would be intractable without the symmetry assumption. If $d$ is *fixed*, and the symmetry of the target $V$ is broken, these critical points persist with spectrum depending continuously on the perturbation of $V$. In summary, we are restricting to a subspace of $M(k,d)$ independent of $d$.  Keeping $k - d$ fixed, and letting $d \rightarrow \infty$ (so $k \rightarrow \infty$), we then prove the existence of an analytic curve of minima emanating from the limiting point pair at infinity. We analyze spectra using tools from group representation theory and analysis

&nbsp;

5. **Extending the analysis to $k-d>2$.**
There is no barrier, but the analysis gets more complex in two ways. Firstly, as we increase $k$, keeping $d$ fixed, new families of critical points appear (emphasis here on families, rather than one family). Secondly, the original critical point is still there but now becomes part of a simplicial complex of degenerate critical points (we call this process fossilization). Once there, the fossilized set remains as we increase $k$ but it never contributes new local minima (the global minimum $V$ becomes part of a connected fossilized set which gives the global minimum).   Each additional neuron increases the dimension of the fixed point space by 2 (for isotropy $S_{d-p}\times S_p,~p > 1$). Generally, as we increase $k-d$, we expect to see more families of critical points. In short, the analysis gets more complex but does not appear so far to be intrinsically different from what we see when $k-d \le 2$. In particular,  fractional power series for minima (or critical points) will exist for new regular families

---

### Meta-Review · Area_Chair_deYS · 2022-08-23

**Recommendation:** Accept
**Confidence:** Less certain

**Metareview:**

Thank you for your submission to NeurIPS.

This paper is on the structure of critical points and local minima in over-parameterized two layer ReLU neural networks.

The reviewers and I, after the author response, are in agreement that there are interesting contributions in this work. However, the reviewers noted significant issues with the presentation (see below). Four knowledgeable reviewers recommend accept/borderline accept, and I concur, in light of the contributions made.

The reviewers also noted several weaknesses in the presentation: In particular, the reviewers noted that
(1) the technical terms used in this paper and notation are hard to follow, which makes the paper not easily accessible
(2) the paper assumes previous knowledge from previous works and is not highly self-readable
(3) the theoretical assumptions are strong,
(4) it is not clear whether the results can give insights on practical neural networks.

Moreover, the analysis techniques are non-standard (to most ML theorist, in my opinion). The reviewers most likely did not check the proofs, but feel confident with the mathematical rigor.

The statement of Theorem 1 looks informal, which should either be made more rigorous or stated that this is an informal version of a formal result that appears later.

Please take into account the updated reviewer comments when preparing the final version to accommodate the requested changes.





**Award:**

No

---

### Decision · Program_Chairs · 2022-09-14

Accept